# FOURIER SLICED-WASSERSTEIN EMBEDDING FOR MULTISETS AND MEASURES

**Tal Amir**
Faculty of Mathematics
Technion–Israel Institute of Technology
Haifa, Israel
`talamir@technion.ac.il`

**Nadav Dym**
Faculty of Mathematics and Faculty of Computer Science
Technion–Israel Institute of Technology
Haifa, Israel
`nadavdym@technion.ac.il`

## ABSTRACT

We present the *Fourier Sliced-Wasserstein (FSW) embedding*—a novel method to embed multisets and measures over $\mathbb{R}^d$ into Euclidean space.

Our proposed embedding approximately preserves the sliced Wasserstein distance on distributions, thereby yielding geometrically meaningful representations that better capture the structure of the input. Moreover, it is injective on measures and *bi-Lipschitz* on multisets—a significant advantage over prevalent methods based on sum- or max-pooling, which are provably not bi-Lipschitz, and, in many cases, not even injective. The required output dimension for these guarantees is near-optimal: roughly $2Nd$, where $N$ is the maximal input multiset size.

Furthermore, we prove that it is *impossible* to embed distributions over $\mathbb{R}^d$ into Euclidean space in a bi-Lipschitz manner. Thus, the metric properties of our embedding are, in a sense, the best possible.

Through numerical experiments, we demonstrate that our method yields superior multiset representations that improve performance in practical learning tasks. Specifically, we show that (a) a simple combination of the FSW embedding with an MLP achieves state-of-the-art performance in learning the (non-sliced) Wasserstein distance; and (b) replacing max-pooling with the FSW embedding makes PointNet significantly more robust to parameter reduction, with only minor performance degradation even after a 40-fold reduction.[1]

## 1 INTRODUCTION

Multisets are unordered collections of vectors that account for repetitions. They are the main mathematical tool for representing unordered data, with perhaps the most notable example being point clouds. As such, there is growing interest in developing architectures suited for learning tasks on multisets. To address this need, several permutation-invariant neural networks have been introduced, with applications for point-cloud classification (Qi et al., 2017a), chemical property prediction (Pozdnyakov & Ceriotti, 2023), and image deblurring (Aittala & Durand, 2018). Multiset aggregation functions also serve as key components in more complex architectures, such as Message Passing Neural Networks (MPNNs) for graphs (Gilmer et al., 2017), or setups with multiple permutation actions (Maron et al., 2020).

A central concept in the study of multiset functions, i.e. functions defined on multisets, is *injectivity*. Its importance is highlighted by the following observation: An architecture that cannot separate two distinct multisets $\boldsymbol{X} \neq \boldsymbol{X}'$ will not be able to approximate a target function $f$ that distinguishes between these multisets, i.e. $f(\boldsymbol{X}) \neq f(\boldsymbol{X}')$. Conversely, a model that maps multisets injectively to vectors, composed with an MLP, can universally approximate *all* continuous multiset functions (Zaheer et al., 2017; Dym & Gortler, 2024). This observation has inspired works to study the injectivity of multiset architectures (Wagstaff et al., 2022; 2019; Tabaghi & Wang, 2024). Injectivity on multisets also plays a key role in the development of expressive MPNNs (Xu et al., 2018).

---

[1]Our code is available at `https://github.com/tal-amir/FSW-embedding`.

Common multiset architectures are typically based on simple building blocks of the form

$$E\left(\{x_1,\ldots,x_n\}\right) = \text{Pool}\Big\{F\Big(\boldsymbol{x}^{(1)}\Big),\ldots,F\Big(\boldsymbol{x}^{(n)}\Big)\Big\},$$

where $F$ is usually an MLP, and Pool is a simple pooling operation such as maximum, sum, or mean. Xu et al. (2018) showed that multiset functions based on max- or mean-pooling are never injective, but injectivity can be achieved using sum-pooling, under the assumption that the vectors $\boldsymbol{x}^{(i)}$ come from a discrete domain, and an appropriate function $F$ is used. Then it was shown by Zaheer et al. (2017); Maron et al. (2019) that injectivity over multisets with continuous elements can be achieved using sum pooling with a polynomial $F$. The more common case, in which $F$ is a neural network, was studied in (Amir et al., 2023). There it was shown that injectivity on multisets and measures over $\mathbb{R}^d$ can be achieved using $F$ that is a shallow MLP with random parameters and analytic, non-polynomial activations, such as Sigmoid or Softplus.

However, injectivity alone is not the strongest property one may desire for multiset functions. While an injective multiset embedding $E$ can separate pairs of distinct multisets $\boldsymbol{X} \neq \boldsymbol{X}'$, this does not ensure the *quality* of the separation. Ideally, if two multisets $\boldsymbol{X}, \boldsymbol{X}'$ are far apart in terms of some notion of distance, then one would expect $E(\boldsymbol{X}), E(\boldsymbol{X}') \in \mathbb{R}^m$ to also be far apart, and vice versa. The standard mathematical notion used to guarantee this behaviour is *bi-Lipschitzness*.

**Definition.** Let $E : \mathcal{D} \to \mathbb{R}^m$, where $\mathcal{D}$ is a collection of multisets or, more generally, distributions over $\mathbb{R}^d$. We say that $E$ is *bi-Lipschitz* with respect to $\mathcal{W}_p$, if there exist constants $c, C > 0$ such that

$$c \cdot \mathcal{W}_p(\mu, \tilde{\mu}) \le \|E(\mu) - E(\tilde{\mu})\| \le C \cdot \mathcal{W}_p(\mu, \tilde{\mu}), \quad \forall \mu, \tilde{\mu} \in \mathcal{D}, \tag{1}$$

where $\mathcal{W}_p$ denotes the $p$-Wasserstein distance and $\| \cdot \|$ denotes the $\ell_2$ norm.

The Wasserstein distance, defined in the next section, serves as a standard notion of distance for multisets and distributions. The ratio of the Lipschitz constants $C/c$ in (1) represents a bound on the maximal distortion incurred by the map $E$, analogous to the condition number of a matrix.

Bi-Lipschitz embeddings enable the application of metric-based learning methods, such as nearest-neighbor search, data clustering, and multi-dimensional scaling, to non-Euclidean data types like multisets and measures. These methods can be more readily applied to the embedded Euclidean domain than to the original data domain, where metric computations are typically more expensive (Indyk & Thaper, 2003). The bi-Lipschitzness of the embedding often provides correctness guarantees for these applications, which depend on the Lipschitz constants $c$ and $C$ (Cahill et al., 2024).

Achieving bi-Lipschitzness is typically more challenging than injectivity and often requires different theoretical tools. Recently, in (Amir et al., 2023), we proved that methods based on average- or sum-pooling can never be bi-Lipschitz, while Xu et al. (2018) showed that methods based on max-pooling cannot even be injective. Thus, a bi-Lipschitz embedding would constitute a substantial improvement over the standard multiset embedding methods currently in use.

To date, two main approaches have been proposed for constructing bi-Lipschitz multiset embeddings: (1) *sort embedding* (Balan et al., 2022), which is based on sorting random projections of the multiset elements, and (2) *max filtering* (Cahill et al., 2022), which is based on computations of Wasserstein distances from 'template multisets', and is computationally expensive.

While sort-based methods have been used with some success (Zhang et al., 2019; 2018; Balan et al., 2022), their popularity in practical applications remains limited, despite their bi-Lipschitzness guarantees. A likely explanation for this is that these methods can only handle multisets of fixed size, and, to date, there is no known way to generalize them to multisets of varying size, let alone to distributions. This is a major limitation, since multisets of varying size arise naturally in many common learning tasks, for example graph classification, where vertices may have neighbourhoods of different sizes. This problem is often circumvented via ad-hoc solutions such as padding (Zhang et al., 2018) or interpolation (Zhang et al., 2019), which do not preserve the original theoretical guarantees of the method. Moreover, even in the restricted setting of fixed-size multisets, the bi-Lipschitzness guarantees of these methods typically require prohibitively high embedding dimensions.

Our goal in this paper is to overcome these limitations by constructing a bi-Lipschitz embedding for the collection of all multisets over $\mathbb{R}^d$ with at most $N$ elements. We denote this collection by $\mathcal{S}_{\le N}(\mathbb{R}^d)$. Note that the assumption of bounded cardinality is necessary, as otherwise, even injectivity is impossible; see (Amir et al., 2023, Theorem C.3). We are also interested in the larger

collection of probability distributions over $\mathbb{R}^d$ supported on at most $N$ points, which we denote by $\mathcal{P}_{\leq N}(\mathbb{R}^d)$. This setting, in which the points may have non-uniform weights, can be particularly relevant for attention-based methods on sets (Lee et al., 2019), as well as graph architectures such as GCN (Kipf & Welling, 2016) or GAT (Veličković et al., 2018), which use non-uniform weights for vertex neighbourhoods. In summary, our main goal is:

**Main goal** For $\mathcal{D} = \mathcal{S}_{\leq N}(\mathbb{R}^d)$ and $\mathcal{D} = \mathcal{P}_{\leq N}(\mathbb{R}^d)$, construct an embedding $E : \mathcal{D} \to \mathbb{R}^m$ that is injective and, preferably, bi-Lipschitz.

**Main results** We propose an embedding method for multisets and distributions with finite support, which is a non-trivial generalization of the sort embedding. We observe that the Euclidean distance between the sort embeddings of two multisets can be interpreted as a finite Monte Carlo sampling of their *sliced Wasserstein* distance (Bonneel et al., 2015): in the special case where the input consists of multisets of fixed size, this sampling corresponds to the project-and-sort operations used in the sort embedding. Based on this interpretation, we extend beyond fixed-size multisets and propose an embedding method for both $\mathcal{S}_{\leq N}(\mathbb{R}^d)$ and $\mathcal{P}_{\leq N}(\mathbb{R}^d)$. Our method essentially operates as follows: (1) compute random one-dimensional projections (also called *slices*) of the input distribution; (2) compute the *quantile function* of each projected distribution; and (3) sample each quantile function at a random frequency in the Fourier domain. We name our method the *Fourier Sliced-Wasserstein (FSW) embedding* and denote it by $E_m^{\mathrm{FSW}}$. The function $E_m^{\mathrm{FSW}} : \mathcal{P}_{\leq N}(\mathbb{R}^d) \to \mathbb{R}^m$ is of the form

$$E_m^{\mathrm{FSW}}(\mu) = E_m^{\mathrm{FSW}}\left(\mu; \left(\boldsymbol{v}^{(k)}, \xi^{(k)}\right)_{k=1}^m\right).$$

It maps multisets and distributions into $\mathbb{R}^m$, and depends on parameters $\boldsymbol{v}^{(k)} \in \mathbb{R}^d$, $\xi^{(k)} \in \mathbb{R}$, $k \in [m]$, representing projection vectors and frequencies respectively. It has the following properties:

1. **Bi-Lipschitzness on multisets:** For $m \geq 2Nd+1$, the map $E_m^{\mathrm{FSW}} : \mathcal{S}_{\leq N}(\mathbb{R}^d) \to \mathbb{R}^m$ is bi-Lipschitz (and, in particular, injective) for almost any choice of the embedding parameters $\left(\boldsymbol{v}^{(k)}, \xi^{(k)}\right)_{k=1}^m$ (Theorem 4.1 and Corollary 4.3).

2. **Injectivity on distributions and measures:** For $m \geq 2Nd + 2N - 1$, the map $E_m^{\mathrm{FSW}} : \mathcal{P}_{\leq N}(\mathbb{R}^d) \to \mathbb{R}^m$ is injective (but not bi-Lipschitz) for almost any choice of parameters (Theorem 4.1). Moreover, by adding one more output coordinate, the embedding can be extended from distributions to measures with arbitrary total mass while preserving injectivity; see Appendix A.1. Furthermore, we prove that it is *impossible* to construct a bi-Lipschitz Euclidean embedding for $\mathcal{P}_{\leq N}(\mathbb{R}^d)$ (Theorem 4.4). This suggests that the metric properties of our embedding are, in a sense, the best possible.

3. **Sliced-Wasserstein approximation:** The expectation of $\frac{1}{m}\left\|E_m^{\mathrm{FSW}}(\mu) - E_m^{\mathrm{FSW}}(\tilde{\mu})\right\|^2$ over the parameters $\left(\boldsymbol{v}^{(k)}, \xi^{(k)}\right)_{k=1}^m$, drawn from our appropriately defined distribution, is exactly the squared sliced-Wasserstein distance between $\mu$ and $\tilde{\mu}$ (Corollary 3.3), with the standard error decreasing as $\mathcal{O}\left(\frac{1}{\sqrt{m}}\right)$.

4. **Differentiability:** The map $E_m^{\mathrm{FSW}}$ is continuous and piecewise smooth in both the input measure parameters $\left(\boldsymbol{x}^{(i)}, w_i\right)_{i=1}^N$ and the embedding parameters $\left(\boldsymbol{v}^{(k)}, \xi^{(k)}\right)_{k=1}^m$ (see e.g. (25)). Thus, it is amenable to gradient-based learning, and its parameters can be trained.

5. **Complexity:** The embedding $E_m^{\mathrm{FSW}}(\mu)$ can be computed efficiently in $\mathcal{O}(mNd + mN\log N)$ time—the same complexity (up to the logarithmic term) as the most efficient methods used in practice (Theorem A.2, Appendix A.2).

In Properties 1 and 2 above, the required embedding dimension $m$ is near-optimal, essentially up to a multiplicative factor of two.

Empirically, in Section 5, we demonstrate the practical promise of our method by evaluating it on the task of learning the (non-sliced) 1-Wasserstein distance function. We show that a simple composition of our embedding with an MLP outperforms the state-of-the-art while requiring shorter training times. Moreover, we show that replacing max-pooling in PointNet with our embedding significantly improves the architecture's robustness in the low-parameter regime, with only minor performance degradation even after a 40-fold parameter reduction.

## 2 PROBLEM SETTING

In this section, we describe the problem in detail and briefly review its theoretical background and existing approaches.

### 2.1 THEORETICAL BACKGROUND

We begin by defining the spaces of multisets and distributions of interest, along with the metrics we use on these spaces.

**Multisets and distributions** Denote by $\mathcal{P}_{\leq N}(\Omega)$ the collection of all probability distributions over a measurable domain $\Omega \subseteq \mathbb{R}^d$, supported on at most $N$ points. Any distribution $\mu \in \mathcal{P}_{\leq N}(\Omega)$ can be parametrized by $N$ points $\boldsymbol{x}^{(i)} \in \Omega$ and weights $w_i \geq 0$ such that

$$\mu = \sum_{i=1}^{N} w_i \delta_{\boldsymbol{x}^{(i)}}, \tag{2}$$

where $\boldsymbol{w} = (w_1, \ldots, w_N)$ is in the probability simplex $\Delta^N \subset \mathbb{R}^N$, and $\delta_{\boldsymbol{x}}$ is Dirac's delta function at $\boldsymbol{x}$. Distributions with fewer than $N$ support points can be parameterized in this way by setting some of the weights $w_i$ to zero and choosing the corresponding $\boldsymbol{x}^{(i)}$ arbitrarily.

Similarly, let $\mathcal{S}_{\leq N}(\Omega)$ be the collection of all multisets over $\Omega \subseteq \mathbb{R}^d$ with at most $N$ elements. To enable computation of Wasserstein distances, which are defined on distributions, we modify the definition of $\mathcal{S}_{\leq N}(\Omega)$ by excluding the empty multiset and identifying each multiset $\boldsymbol{X} = \left\{\boldsymbol{x}^{(i)}\right\}_{i=1}^{n} \in \mathcal{S}_{\leq N}(\Omega)$ with the distribution $\mu[\boldsymbol{X}] \in \mathcal{P}_{\leq N}(\Omega)$ that assigns uniform weights $w_i = \frac{1}{n}$ to all $\boldsymbol{x}^{(i)}$, taking their multiplicities into account.[2] Throughout this work, our use of $\mathcal{S}_{\leq N}(\Omega)$ assumes this modification, and we regard $\mathcal{S}_{\leq N}(\Omega)$ as a subset of $\mathcal{P}_{\leq N}(\Omega)$.

Note that our embedding in its basic form, to be defined in the next section, will take input multisets from $\mathcal{S}_{\leq N}(\Omega)$. As such, it will not be defined on the empty multiset, and will not distinguish between multisets of different sizes if their element proportions are identical.[3] This can be easily remedied by augmenting the embedding with an additional output coordinate that encodes the multiset cardinality, or in the case of measures, the *total mass* $\sum_{i=1}^{N} w_i$; see Appendix A.1 for details.

Throughout this work, we focus on $\Omega = \mathbb{R}^d$ and only discuss finitely-supported multisets and distributions. Nonetheless, our embedding can accommodate general distributions over $\mathbb{R}^d$, while retaining its sliced-Wasserstein approximation property. Thus, in principle, our method can be applied to structures other than point clouds, for example, polygonal meshes and volumetric data.

**Wasserstein distance** As a distance measure on $\mathcal{S}_{\leq N}(\mathbb{R}^d)$ and $\mathcal{P}_{\leq N}(\mathbb{R}^d)$, we use the aforementioned Wasserstein distance. Intuitively, this distance is the least amount of work required to 'transport' one distribution to another. For two distributions $\mu, \tilde{\mu} \in \mathcal{P}_{\leq N}(\mathbb{R}^d)$, parametrized by points $\boldsymbol{x}^{(i)}, \tilde{\boldsymbol{x}}^{(j)}$ and weights $w_i, \tilde{w}_j$ as in (2), the $p$-Wasserstein distance between $\mu$ and $\tilde{\mu}$ is defined by

$$\mathcal{W}_p(\mu, \tilde{\mu}) := \left( \inf_{\pi \in \Pi(\mu, \tilde{\mu})} \sum_{i,j \in [N]} \pi_{ij} \left\| \boldsymbol{x}^{(i)} - \tilde{\boldsymbol{x}}^{(j)} \right\|^p \right)^{\frac{1}{p}} \qquad p \in [1, \infty),$$

$$\mathcal{W}_\infty(\mu, \tilde{\mu}) := \inf_{\pi \in \Pi(\mu, \tilde{\mu})} \max \left\{ \left\| \boldsymbol{x}^{(i)} - \tilde{\boldsymbol{x}}^{(j)} \right\| \ \middle| \ i, j \in [N], \pi_{ij} > 0 \right\},$$

where $\|\cdot\|$ is the Euclidean norm, and $\Pi(\mu, \tilde{\mu})$ is the set of all *transport plans* from $\mu$ to $\tilde{\mu}$:

$$\Pi(\mu, \tilde{\mu}) := \left\{ \pi \in \mathbb{R}^{N \times N} \ \middle| \ (\forall i, j \in [N]) \ \pi_{ij} \geq 0 \wedge \sum_{j \in [N]} \pi_{ij} = w_i \wedge \sum_{i \in [N]} \pi_{ij} = \tilde{w}_j \right\}.$$

---

[2] For example, the multiset $\boldsymbol{X} = \{a, b, b\} \in \mathcal{S}_{\leq 3}(\mathbb{R})$ is identified with the distribution $\mu[\boldsymbol{X}] = \frac{1}{3}\delta_a + \frac{2}{3}\delta_b \in \mathcal{P}_{\leq 3}(\mathbb{R})$.

[3] e.g., $\boldsymbol{X} = \{a, b, b\}$ and $\boldsymbol{Y} = \{a, a, b, b, b, b\}$ are considered identical in $\mathcal{S}_{\leq 6}(\mathbb{R})$, since $\mu[\boldsymbol{X}] = \mu[\boldsymbol{Y}]$.

The numbers $\pi_{ij}$ intuitively represent the amount of mass to be transported from point $\boldsymbol{x}^{(i)}$ to $\tilde{\boldsymbol{x}}^{(j)}$. Whenever $p$ is omitted, we refer to $\mathcal{W}_p$ with $p = 2$. Similarly, $\|\cdot\|$ always denotes the $\ell_2$ norm.

**Computation of Wasserstein**  The Wasserstein distance can be computed in $\mathcal{O}(N^3 \log N)$ time via a linear program (Altschuler et al., 2017; Orlin, 1988). Alternatively, one may use the Sinkhorn approximation algorithm (Cuturi, 2013), which takes $\tilde{\mathcal{O}}(N^2\varepsilon^{-3})$ time, $\varepsilon$ being the error tolerance (Altschuler et al., 2017). This was improved to $\tilde{\mathcal{O}}(\min\{N^{2.25}\varepsilon^{-1}, N^2\varepsilon^{-2}\})$ in (Dvurechensky et al., 2018). However, in the special case $d = 1$, it can be computed significantly faster.

**Wasserstein in 1D**  In one dimension, the Wasserstein distance can be computed in only $\mathcal{O}(N \log N)$ time. If $\boldsymbol{x} = (x_1, \ldots, x_n)$, $\boldsymbol{y} = (y_1, \ldots, y_n)$ are two vectors in $\mathbb{R}^n$, then the distance between the two uniform distributions $\mu[\boldsymbol{x}], \mu[\boldsymbol{y}]$ over the vector coordinates is given by

$$\mathcal{W}(\mu[\boldsymbol{x}], \mu[\boldsymbol{y}]) = \frac{1}{\sqrt{n}}\|\text{sort}(\boldsymbol{x}) - \text{sort}(\boldsymbol{y})\|, \tag{3}$$

with sort $: \mathbb{R}^n \to \mathbb{R}^n$ being the function that returns the input coordinates sorted in increasing order.

For arbitrary distributions in $\mathcal{P}_{\leq N}(\mathbb{R})$, the Wasserstein distance can be computed similarly via the *quantile function*. For a distribution $\mu$ over $\mathbb{R}$, the quantile function $Q_\mu : [0, 1) \to \mathbb{R}$ is a continuous analogue of the sort function, defined by

$$Q_\mu(t) := \inf\{x \in \mathbb{R} \mid \mu((-\infty, x]) > t\}.$$

Figure 1 depicts the quantile functions for three different multisets.

The quantile function enables an explicit formula for the Wasserstein distance between two distributions over $\mathbb{R}$ (see e.g. Bayraktar & Guo (2021), Eq. 2.3 and the paragraph thereafter):

$$\mathcal{W}(\mu, \tilde{\mu}) = \sqrt{\int_0^1 (Q_\mu(t) - Q_{\tilde{\mu}}(t))^2 dt}. \tag{4}$$

Note that when $\mu$ and $\tilde{\mu}$ are generated by multisets of the same cardinality (like the two multisets of cardinality 3 in Figure 1), the formulas (4) and (3) coincide.

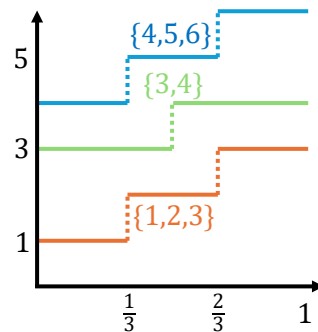

Figure 1: Quantile functions for three different multisets

**Sliced Wasserstein distance**  The *sliced Wasserstein distance*, proposed by Bonneel et al. (2015) as a surrogate for the Wasserstein distance, exploits the efficient computation of the latter for $d = 1$ to define a more computationally tractable distance for $d > 1$. It is defined as the average Wasserstein distance between all one-dimensional projections (or *slices*) of the two input distributions. For a formal definition, we first define the projection of a distribution.

**Definition.**  Let $\mu = \sum_{i=1}^N w_i \delta_{\boldsymbol{x}^{(i)}} \in \mathcal{P}_{\leq N}(\mathbb{R}^d)$. The *projection* of $\mu$ in the direction $\boldsymbol{v} \in \mathbb{R}^d$, denoted by $\boldsymbol{v}^T\mu$, is the one-dimensional distribution in $\mathcal{P}_{\leq N}(\mathbb{R})$ defined by

$$\boldsymbol{v}^T\mu := \sum_{i=1}^N w_i \delta_{\boldsymbol{v}^T\boldsymbol{x}^{(i)}}.$$

Using the above definition, the sliced Wasserstein distance between $\mu, \tilde{\mu} \in \mathcal{P}_{\leq N}(\mathbb{R}^d)$ is defined by

$$\mathcal{SW}(\mu, \tilde{\mu}) := \sqrt{\mathbb{E}_{\boldsymbol{v}}[\mathcal{W}^2(\boldsymbol{v}^T\mu, \boldsymbol{v}^T\tilde{\mu})]}, \tag{5}$$

where $\mathcal{W}^2(\cdot, \cdot)$ is the squared 2-Wasserstein distance, and the expectation $\mathbb{E}_{\boldsymbol{v}}[\cdot]$ is over the direction vector $\boldsymbol{v} \sim \text{Uniform}(\mathbb{S}^{d-1})$, i.e. distributed uniformly over the unit sphere in $\mathbb{R}^d$.

## 2.2 Existing embedding methods

We now return to our main goal of constructing an embedding $E : \mathcal{P}_{\leq N}(\mathbb{R}^d) \to \mathbb{R}^m$. In this subsection, we discuss existing embedding methods and some straightforward ideas to extend them.

We first observe that, on the collection of multisets over $\mathbb{R}$ of fixed size $n$, it follows from (3) that the map $\{x_1, \ldots, x_n\} \mapsto \frac{1}{\sqrt{n}} \cdot \text{sort}(x_1, \ldots, x_n)$ is an isometry, i.e. (1) holds with $C = c = 1$.

To extend this idea to multisets in $\mathcal{S}_{\leq N}(\mathbb{R})$, a naive approach would be to represent each multiset in $\mathcal{S}_{\leq N}(\mathbb{R})$ by a multiset of size $\hat{N}$, with $\hat{N}$ being the *least common multiple* (LCM) of $\{1, 2, \ldots, N\}$. For example, for $N = 3$, LCM($\{1, 2, 3\}$) $= 6$, and thus multisets in $\mathcal{S}_{\leq N}(\mathbb{R})$ of sizes 1 $\{a\}$, 2 $\{a, b\}$, and 3 $\{a, b, c\}$, would be represented, respectively, by $\{a, a, a, a, a, a\}$, $\{a, a, a, b, b, b\}$ and $\{a, a, b, b, c, c\}$. At this point, a sorting approach can be applied. However, as $N$ increases, this method quickly becomes infeasible, both in terms of computation time as well as memory, since $\hat{N}$ grows exponentially in $N$. Moreover, this method cannot handle arbitrary distributions in $\mathcal{P}_{\leq N}(\mathbb{R})$, whose weights may be irrational.

One possible approach to embed general distributions $\mu \in \mathcal{P}_{\leq N}(\mathbb{R})$ is to sample $Q_\mu(t)$ at $m$ points $t_1, \ldots, t_m \in [0, 1]$ equispaced or drawn uniformly at random. While this approach would indeed approximately preserve the Wasserstein distance, as follows from (4), it is easy to show that for any finite number of samples $m$, this embedding is not injective on $\mathcal{P}_{\leq N}(\mathbb{R})$. Moreover, it is discontinuous with respect to the sampling points $t_k$ and input probabilities $w_i$, and thus not amenable to gradient-based learning methods. Our method, described in the next section, resolves these issues by sampling the quantile function in the frequency domain rather than in the $t$-domain.

When considering $\mathcal{P}_{\leq N}(\mathbb{R}^d)$ with $d > 1$, one natural idea is to take $m$ one-dimensional projections of the input distribution, and then embed each of the projections using one of the methods described above for $\mathcal{P}_{\leq N}(\mathbb{R})$. In the case of multisets of fixed cardinality $n$, this corresponds to the mapping

$$\{\boldsymbol{x}_1, \ldots, \boldsymbol{x}_n\} \mapsto \frac{1}{\sqrt{mn}} \cdot \text{rowsort}\left(\left[\boldsymbol{v}_k^T \boldsymbol{x}_i\right]_{k \in [m], i \in [n]}\right).$$

This idea was discussed in (Balan et al., 2022; Zhang et al., 2019; Dym & Gortler, 2024; Balan & Tsoukanis, 2023b). In expectation over the directions $\boldsymbol{v}_k$, this method indeed gives a good approximation of the sliced Wasserstein distance. The relationship to the $d$-dimensional Wasserstein distance is *a priori* less clear. Balan & Tsoukanis (2023a) showed that for $m$ that is exponential in $n$, this mapping bi-Lipschitz for almost any choice of $\boldsymbol{v}_1, \ldots, \boldsymbol{v}_m$. Later, Dym & Gortler (2024) showed that $m = 2nd + 1$ is sufficient. In this paper, we combine this idea of using linear projections, with our idea of Fourier sampling of quantile functions, to construct an embedding defined on the whole of $\mathcal{P}_{\leq N}(\mathbb{R}^d)$ while maintaining theoretical guarantees and practical efficiency.

In a related line of work, Kolouri et al. (2015); Naderializadeh et al. (2021); Lu et al. (2024) developed a method that embeds continuous distributions into an infinite-dimensional Hilbert space isometrically with respect to the sliced Wasserstein distance. However, in practice, a finite-dimensional discretization is used, which breaks the method's injectivity, as we show empirically in Section 5. In contrast, our method is provably injective with a finite and near-optimal embedding dimension of $\approx 2Nd$.

Lastly, Haviv et al. (2024) recently proposed a transformer-based architecture called Wasserstein Wormhole, designed to compute Euclidean embeddings for multisets and distributions. This architecture can be trained to approximate the Wasserstein distance. However, its reliance on training is a key limitation, as it does not guarantee Wasserstein distance preservation or even injectivity. This concern becomes particularly significant when generalizing to out-of-distribution samples.

## 3 PROPOSED METHOD

Our method for embedding a distribution $\mu$ essentially consists of computing random slices $\boldsymbol{v}^T \mu$ and, for each slice, taking a single random sample of its quantile function $Q_{\boldsymbol{v}^T \mu}(t)$. However, instead of sampling the function directly, we sample its *cosine transform*—a variant of the Fourier transform. Since the Fourier transform is a linear isometry, integrating the squared difference of these samples for two distributions $\mu, \tilde{\mu}$ yields the squared sliced Wasserstein distance $\mathcal{SW}^2(\mu, \tilde{\mu})$, as we shall show next. We will also show that this sampling method guarantees injectivity, unlike direct sampling of $Q_{\boldsymbol{v}^T \mu}(t)$. Lastly, since $Q_{\boldsymbol{v}^T \mu}(t)$ is a compactly supported step function, its Fourier transform is smooth with respect to the frequencies, and thus so is our embedding. We now discuss this in detail.

**Definition 3.1.** Given a *projection vector* $\boldsymbol{v} \in \mathbb{S}^{d-1}$ and a number $\xi \geq 0$ denoting a frequency, we define the *one-sample embedding* $E^{\text{FSW}}(\,\cdot\,; \boldsymbol{v}, \xi) : \mathcal{P}_{\leq N}(\mathbb{R}^d) \to \mathbb{R}$ by

$$E^{\text{FSW}}(\mu; \boldsymbol{v}, \xi) := 2(1 + \xi) \int_0^1 Q_{\boldsymbol{v}^T \mu}(t) \cos(2\pi \xi t) dt, \tag{6}$$

which is the *cosine transform* of $Q_{\boldsymbol{v}^T \mu}(t)$, sampled at frequency $\xi$ and multiplied by $1 + \xi$; see Appendix B.1 for further discussion. Details on the practical computation of $E^{\text{FSW}}$ are in Appendix A.2.

Next, we define a probability distribution $\mathcal{D}_\xi$ for the frequency $\xi$, given by the PDF

$$f_\xi(\xi) := (1 + \xi)^{-2}, \qquad \xi \geq 0.$$

We now show that $E^{\text{FSW}}$ preserves the sliced Wasserstein distance in expectation over $\boldsymbol{v}, \xi$.

**Theorem 3.2.** (Proof in Appendix B.2) *Let* $\mu, \tilde{\mu} \in \mathcal{P}_{\leq N}(B_R)$, *with* $B_R \subset \mathbb{R}^d$ *being the closed* $\ell_2$-*norm ball of radius* $R$, *centered at zero. Let* $\boldsymbol{v} \sim \text{Uniform}(\mathbb{S}^{d-1})$, $\xi \sim \mathcal{D}_\xi$. *Then*

$$\mathbb{E}_{\boldsymbol{v}, \xi}\left[ \left| E^{\text{FSW}}(\mu; \boldsymbol{v}, \xi) - E^{\text{FSW}}(\tilde{\mu}; \boldsymbol{v}, \xi) \right|^2 \right] = \mathcal{SW}^2(\mu, \tilde{\mu}), \tag{7}$$

$$\text{STD}_{\boldsymbol{v}, \xi}\left[ \left| E^{\text{FSW}}(\mu; \boldsymbol{v}, \xi) - E^{\text{FSW}}(\tilde{\mu}; \boldsymbol{v}, \xi) \right|^2 \right] \leq 13 \cdot R^2. \tag{8}$$

This result can be further stabilized by taking multiple samples. Thus, we define the *Fourier Sliced Wasserstein (FSW) embedding* $E_m^{\text{FSW}} : \mathcal{P}_{\leq N}(\mathbb{R}^d) \to \mathbb{R}^m$ by

$$E_m^{\text{FSW}}(\mu) := \left( E^{\text{FSW}}(\mu; \boldsymbol{v}^{(1)}, \xi^{(1)}), \ldots, E^{\text{FSW}}(\mu; \boldsymbol{v}^{(m)}, \xi^{(m)}) \right), \tag{9}$$

where $\left( \boldsymbol{v}^{(k)}, \xi^{(k)} \right)_{k=1}^m$ are drawn i.i.d. from $\text{Uniform}(\mathbb{S}^{d-1}) \times \mathcal{D}_\xi$. Consequently, we have:

**Corollary 3.3.** *Under the assumptions of Theorem 3.2,*

$$\mathbb{E}_{\boldsymbol{v}, \xi}\left[ \tfrac{1}{m} \left\| E_m^{\text{FSW}}(\mu) - E_m^{\text{FSW}}(\tilde{\mu}) \right\|^2 \right] = \mathcal{SW}^2(\mu, \tilde{\mu}), \tag{10}$$

$$\text{STD}_{\boldsymbol{v}, \xi}\left[ \tfrac{1}{m} \left\| E_m^{\text{FSW}}(\mu) - E_m^{\text{FSW}}(\tilde{\mu}) \right\|^2 \right] \leq 13 \cdot \frac{R^2}{\sqrt{m}}. \tag{11}$$

Notably, the error bound in (11) is independent of the number of points $N$ and the dimension $d$. Thus, the estimation error does not suffer from the curse of dimensionality. By taking a sufficiently high embedding dimension, one can embed distributions of arbitrarily high dimension with arbitrary (and possibly infinite) support cardinality, while ensuring a bounded standard estimation error, as long as the distributions are supported within a fixed ball of radius $R$.

## 4 THEORETICAL RESULTS

In the previous section, we showed that our embedding approximately preserves the sliced Wasserstein distance in a probabilistic sense, with diminishing estimation error as the embedding dimension grows. Here, we show that with a *finite* dimension, our embedding is guaranteed to be injective and bi-Lipschitz, as outlined in the Main Results summary in Section 1.

First, we show that with a sufficiently high dimension $m$, our embedding is injective.

**Theorem 4.1.** (Proof in Appendix B.3) *Let* $E_m^{\text{FSW}} : \mathcal{P}_{\leq N}(\mathbb{R}^d) \to \mathbb{R}^m$ *be as in (9), with* $\left( \boldsymbol{v}^{(k)}, \xi^{(k)} \right)_{k=1}^m$ *drawn i.i.d. from* $\text{Uniform}(\mathbb{S}^{d-1}) \times \mathcal{D}_\xi$. *Then:*

1. *If* $m \geq 2Nd + 1$, *then with probability 1,* $E_m^{\text{FSW}}$ *is injective on* $\mathcal{S}_{\leq N}(\mathbb{R}^d)$.

2. *If* $m \geq 2Nd + 2N - 1$, *then with probability 1,* $E_m^{\text{FSW}}$ *is injective on* $\mathcal{P}_{\leq N}(\mathbb{R}^d)$.

These bounds are optimal essentially up to a multiplicative factor of 2, since an embedding dimension $m < Nd$ precludes injectivity for *any* continuous embedding (Amir et al., 2023, Theorem C.3).

Next, we show that in the case of $\mathcal{S}_{\leq N}(\mathbb{R}^d)$, the injectivity of $E_m^{\text{FSW}}$ implies that it is, in fact, bi-Lipschitz. Our proof relies on the fact that $E_m^{\text{FSW}}$ is piecewise linear and homogeneous in the input points, in a sense we shall now define. By a slight abuse of notation, we denote the distribution parametrized by the points $\boldsymbol{X} = \left( \boldsymbol{x}^{(1)}, \ldots, \boldsymbol{x}^{(N)} \right)$ and weights $\boldsymbol{w} = (w_1, \ldots, w_N)$ as $(\boldsymbol{X}, \boldsymbol{w})$.

**Definition.** Let $E : \mathcal{D} \to \mathbb{R}^m$ with $\mathcal{D} = \mathcal{P}_{\leq N}(\mathbb{R}^d)$ or $\mathcal{S}_{\leq N}(\mathbb{R}^d)$. We say that $E$ is *positively homogeneous* if for any $\alpha \geq 0$ and any distribution $(\boldsymbol{X}, \boldsymbol{w}) \in \mathcal{D}$, $E(\alpha \boldsymbol{X}, \boldsymbol{w}) = \alpha E(\boldsymbol{X}, \boldsymbol{w})$.

The next theorem shows that any injective embedding of $\mathcal{P}_{\leq N}(\mathbb{R}^d)$ that is positively homogeneous and piecewise linear[4] is bi-Lipschitz when restricted to distributions with fixed weights.

**Theorem 4.2.** (Proof in Appendix B.3) *Let $E : \mathcal{P}_{\leq N}(\mathbb{R}^d) \to \mathbb{R}^m$ be injective and positively homogeneous. Let $\boldsymbol{w}, \tilde{\boldsymbol{w}} \in \Delta^N$ be fixed, and suppose that the functions $E(\boldsymbol{X}, \boldsymbol{w})$ and $E(\boldsymbol{X}, \tilde{\boldsymbol{w}})$ are piecewise linear in $\boldsymbol{X}$. Then there exist $C, c > 0$ such that for all $\boldsymbol{X}, \tilde{\boldsymbol{X}} \in \mathbb{R}^{d \times N}$ and $p \in [1, \infty]$,*

$$c \cdot \mathcal{W}_p\Big((\boldsymbol{X}, \boldsymbol{w}), \big(\tilde{\boldsymbol{X}}, \tilde{\boldsymbol{w}}\big)\Big) \leq \Big\| E(\boldsymbol{X}, \boldsymbol{w}) - E\big(\tilde{\boldsymbol{X}}, \tilde{\boldsymbol{w}}\big)\Big\| \leq C \cdot \mathcal{W}_p\Big((\boldsymbol{X}, \boldsymbol{w}), \big(\tilde{\boldsymbol{X}}, \tilde{\boldsymbol{w}}\big)\Big). \quad (12)$$

The restriction to fixed weights can easily be relaxed to weights from any finite set. Based on this observation, we now show that $E_m^{\mathrm{FSW}}$ is bi-Lipschitz on multisets.

**Corollary 4.3.** *Let $E_m^{\mathrm{FSW}}$ be as in* (9) *with $m \geq 2Nd + 1$. Then with probability 1, $E_m^{\mathrm{FSW}}$ is bi-Lipschitz on $\mathcal{S}_{\leq N}(\mathbb{R}^d)$.*

*Proof.* Any $\mu \in \mathcal{S}_{\leq N}(\mathbb{R}^d)$ can be represented by a parameter of the form $(\boldsymbol{X}, \boldsymbol{w}^{(n)})$, where

$$\boldsymbol{w}^{(n)} = \Big(\overbrace{\tfrac{1}{n}, \ldots, \tfrac{1}{n}}^{n}, \overbrace{0, \ldots, 0}^{N-n}\Big), \qquad 1 \leq n \leq N. \quad (13)$$

For $n, r \in [N]$, let $c_{nr}, C_{nr}$ be the Lipschitz constants $c, C$ of (12) for $E_m^{\mathrm{FSW}}$ with the weight vectors $\boldsymbol{w} = \boldsymbol{w}^{(n)}, \tilde{\boldsymbol{w}} = \boldsymbol{w}^{(r)}$. Then, it follows directly that $E_m^{\mathrm{FSW}}$ is bi-Lipschitz on $\mathcal{S}_{\leq N}(\mathbb{R}^d)$ with the constants $c = \min_{n,r \in [N]} c_{nr} > 0$ and $C = \max_{n,r \in [N]} C_{nr} < \infty$. □

Next, we explore whether it is possible to further improve by finding a bi-Lipschitz embedding for the whole of $\mathcal{P}_{\leq N}(\mathbb{R}^d)$. For the broader class of distributions $\bigcup_{N \in \mathbb{N}} \mathcal{P}_{\leq N}(\mathbb{R}^d)$, Naor & Schechtman (2007) proved that no bi-Lipschitz embedding exists into $L^1([0, 1])$, and thus not into any finite-dimensional space. One may ask whether this impossibility can be circumvented by bounding the number of support points and confining them to a fixed compact domain, namely, by restricting to $\mathcal{P}_{\leq N}(\Omega)$ for a compact $\Omega \subset \mathbb{R}^d$. The following theorem shows that even this restricted collection still cannot be embedded in a bi-Lipschitz manner into any finite-dimensional Euclidean space.

**Theorem 4.4.** (Proof in Appendix B.3) *Let $E : \mathcal{P}_{\leq N}(\Omega) \to \mathbb{R}^m$, where $N \geq 2$ and $\Omega \subseteq \mathbb{R}^d$ has a nonempty interior. Then for all $p \in [1, \infty]$, $E$ is not bi-Lipschitz on $\mathcal{P}_{\leq N}(\Omega)$ with respect to $\mathcal{W}_p$.*

## 5 NUMERICAL EXPERIMENTS

In this section, we demonstrate how the theoretical strengths of our method manifest in practice. Specifically, we show that our method produces embeddings with superior distance preservation and improves performance in practical learning tasks.

**Comparison with PSWE** This experiment compares our method with PSWE (Naderializadeh et al., 2021)—an embedding method developed with the same purpose of preserving the sliced Wasserstein distance. As mentioned in Section 2.2, this method embeds continuous distributions into an infinite-dimensional Hilbert space, and does so isometrically with respect to the sliced Wasserstein distance. Finite multisets and distributions are treated by this method as discretizations of continuous distributions, which causes different finite distributions to be identified with the same continuous distribution, consequently leading to a loss of injectivity.

To demonstrate this, we constructed a pair of multisets $\boldsymbol{X}_1, \boldsymbol{X}_2 \subset \mathbb{R}^3$, of sizes 5 and 200, respectively, that PSWE identifies as the same continuous distribution, and thus fails to distinguish; see Appendix C.1 for details. Table 1 reports the normalized Euclidean distances $\frac{1}{\sqrt{m}}\|E(\boldsymbol{X}_1) - E(\boldsymbol{X}_2)\|$, where $E : \mathcal{S}_{\leq N}(\mathbb{R}^3) \to \mathbb{R}^m$ is either FSW or PSWE, for various embedding dimensions $m$. As shown in the table, PSWE fails to distinguish the two multisets for all tested values of $m$, whereas FSW succeeds even when $m = 1$. Moreover, as $m$ increases, the distances produced by FSW converge to the sliced Wasserstein distance $\mathcal{SW}(\boldsymbol{X}_1, \boldsymbol{X}_2)$, as promised by Corollary 3.3.

---

[4]A definition of piecewise linearity appears in Definition B.12 Appendix B.3.

Table 1: Embedding distance vs. embedding dimension

| Method | Embedding dimension | | | | |
|---|---|---|---|---|---|
| | 1 | 10 | 100 | 1000 | 10000 |
| FSW | 0.125 | 0.0326 | 0.0639 | 0.0733 | 0.0737 |
| PSWE | 0.0 | 5.48e-8 | 1.47e-7 | 1.3e-7 | 1.44e-7 |

Embedding distances between a pair of multisets $\boldsymbol{X}_1, \boldsymbol{X}_2 \subset \mathbb{R}^3$ distinguished by FSW but not by PSWE. Here $|\boldsymbol{X}_1| = 5$, $|\boldsymbol{X}_2| = 200$, and $\mathcal{SW}(\boldsymbol{X}_1, \boldsymbol{X}_2) = 0.0734$.

**Learning to approximate the Wasserstein distance**  One possible approach to overcome the high computation time of the Wasserstein distance for $d > 1$ is to try to estimate it using a neural network, trained on pairs of point-clouds for which the distance is known. This approach was used in previous works (Chen & Wang, 2023; Kawano et al., 2020), which proposed architectures designed to approximate functions $F : \mathcal{S}_{\leq N}(\mathbb{R}^d) \times \mathcal{S}_{\leq N}(\mathbb{R}^d) \to \mathbb{R}$, such as the Wasserstein distance function. These methods handle multisets using the traditional approach of sum- or average-pooling. Since our embedding is bi-Lipschitz with respect to the Wasserstein distance, it is *a priori* likely to be a more effective building block for architectures designed to learn it.

For this task, we used the following architecture: First, an FSW embedding $E_1 : \mathcal{P}_{\leq N}(\mathbb{R}^d) \to \mathbb{R}^{m_1}$ is applied to each of the two input distributions $\mu, \tilde{\mu}$. Then, a second FSW embedding $E_2 : \mathcal{S}_{\leq 2}(\mathbb{R}^{m_1}) \to \mathbb{R}^{m_2}$ is applied to the multiset $\{E_1(\mu), E_1(\tilde{\mu})\}$. The output of $E_2$ is then fed to an MLP $\Phi : \mathbb{R}^{m_2} \to \mathbb{R}_+$; see Appendix C.2 for dimensions and technical details. Our full architecture is described by the formula

$$F(\mu, \tilde{\mu}) := \Phi(E_2(\{E_1(\mu), E_1(\tilde{\mu})\})).$$

This formulation ensures that $F$ is symmetric with respect to swapping $\mu$ and $\tilde{\mu}$. In addition, we used leaky-ReLU activations and no biases in $\Phi$, which renders $F$ scale-equivariant by design, i.e.

$$F\big((\alpha \boldsymbol{X}, \boldsymbol{w}), (\alpha \tilde{\boldsymbol{X}}, \tilde{\boldsymbol{w}})\big) = \alpha F\big((\boldsymbol{X}, \boldsymbol{w}), (\tilde{\boldsymbol{X}}, \tilde{\boldsymbol{w}})\big) \quad \forall \alpha > 0,$$

as is the Wasserstein distance function, which $F$ is designed to approximate.

The experimental setting was replicated from (Chen & Wang, 2023). The objective is to approximate the 1-Wasserstein distance $\mathcal{W}_1$. The following evaluation datasets, kindly provided to us by the authors, were used: Three synthetic datasets `noisy-sphere-3`, `noisy-sphere-6` and `uniform`, with random point-clouds in $\mathbb{R}^3$, $\mathbb{R}^6$ and $\mathbb{R}^2$; two real datasets `ModelNet-small` and `ModelNet-large`, with 3D point-clouds sampled from ModelNet40 objects (Wu et al., 2015); and the gene-expression dataset `RNAseq` (Yao et al., 2021), with multisets in $\mathbb{R}^{2000}$.

We compared our architecture to the following methods: (a) $\mathcal{N}_{\text{SDeepSets}}$, a DeepSets-like architecture trained to compute $\mathcal{W}_1$-preserving embeddings, and $\mathcal{N}_{\text{ProductNet}}$, which further processes the sum of the two embeddings through an MLP (Chen & Wang, 2023); (b) a Siamese autoencoder called Wasserstein Point-Cloud Embedding network (WPCE) (Kawano et al., 2020); and (c) the Sinkhorn approximation algorithm (Cuturi, 2013). We also evaluated the PSWE embedding by employing it in our architecture instead of $E_1$.

Table 2: 1-Wasserstein approximation: Relative error

| Dataset | $d$ | set size | Ours | PSWE | $\mathcal{N}_{\text{ProductNet}}$ | WPCE | $\mathcal{N}_{\text{SDeepSets}}$ | Sinkhorn |
|---|---|---|---|---|---|---|---|---|
| noisy-sphere-3 | 3 | 100–299 | **1.4 %** | 2.2 % | 4.6 % | 34.1 % | 36.2 % | 18.7 % |
| noisy-sphere-6 | 6 | 100–299 | **1.3 %** | 1.4 % | 1.5 % | 26.9 % | 29.1 % | 13.7 % |
| uniform | 2 | 256 | 2.4 % | **2.1 %** | 9.7 % | 12.0 % | 12.3 % | 7.3 % |
| ModelNet-small | 3 | 20–199 | **2.9 %** | 5.7 % | 8.4 % | 7.7 % | 10.5 % | 10.1 % |
| ModelNet-large | 3 | 2047 | 2.6 % | **2.4 %** | 14.0 % | 15.9 % | 16.6 % | 14.8 % |
| RNAseq | 2000 | 20–199 | **1.1 %** | 1.2 % | 1.2 % | 47.7 % | 48.2 % | 4.0 % |

Mean relative error in approximating the 1-Wasserstein distance between point sets.

As seen in Table 2, our architecture achieves the best accuracy on most evaluation datasets. Training times are in Table 3. Further details appear in Appendix C.2.

Table 3: 1-Wasserstein approximation: Training time

| Dataset | Ours | PSWE | $\mathcal{N}_{\text{ProductNet}}$ | WPCE | $\mathcal{N}_{\text{SDeepSets}}$ |
|---|---|---|---|---|---|
| noisy-sphere-3 | 2.2 min | 33 min | 6 min | 1 h 46 min | 9 min |
| noisy-sphere-6 | 4 min | 1 h | 12 min | 4 h 6 min | 1 h 38 min |
| uniform | 3 min | 51 min | 7 min | 3 h 36 min | 1 h 27 min |
| ModelNet-small | 3 min | 48 min | 7 min | 1 h 23 min | 12 min |
| ModelNet-large | 14.2 min | 1 h 19 min | 8 min | 3 h 5 min | 40 min |
| RNAseq | 4 min | 50 min | 15 min | 14 h 26 min | 3 h 1 min |

Training times for the different architectures.

**Robustness to parameter reduction**  Next, we evaluate our embedding on the `ModelNet40` object classification benchmark (Wu et al., 2015). This dataset consists of 3D point clouds representing objects coming from 40 different classes.

To demonstrate the improved representational power of our method, we used a simple architecture of the form $F(\mu) = \Phi(E(\mu))$, where $E : \mathcal{S}_{\leq N}(\mathbb{R}^d) \to \mathbb{R}^m$ is the FSW embedding, $\Phi : \mathbb{R}^m \to \mathbb{R}^C$ is an MLP and $C = 40$ is the number of classes. We compared this architecture with PointNet (Qi et al., 2017a), which applies an MLP elementwise, followed by max pooling and another MLP. We evaluated different sizes of both architectures to assess their robustness to parameter reduction.

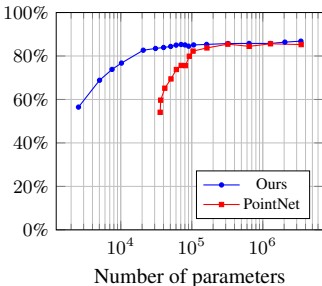

Figure 2: ModelNet40 classification accuracy

With 300 k parameters or more, the two methods perform comparably: our model achieves 85.7 %–86.8 % accuracy and PointNet[5] achieves 84.4 %–85.6 %. However, our model is considerably more robust with fewer parameters, as shown in Figure 2. For example, with 31 k parameters, our model achieves 83.5 % accuracy, whereas PointNet achieves only 54.1 % with 35 k parameters.

We note that simple methods such as PointNet and ours do not achieve the best results on ModelNet40. More complex methods achieve superior results by applying PointNet-based embeddings to local neighborhoods of each point. For example, PointNet++ (Qi et al., 2017b) achieved 90.7 % accuracy, and (Mohammadi et al., 2021) achieved an impressive 95.4 %. Based on our preliminary results, we believe that combining our embedding with such methods, by employing it in their multiset aggregation steps, may enhance robustness to parameter reduction. This property can be crucial in practical applications.

## 6  CONCLUSION

In this paper, we introduced the first injective embedding for multisets and measures that is provably bi-Lipschitz on multisets. Our embedding is computationally efficient, and both theoretical and empirical results indicate that it better preserves the original geometry of the data, consequently leading to improved performance in practical learning tasks.

Our embedding has already shown promising results in subsequent work (Sverdlov et al., 2024), where it was used to construct the first graph neural network with bi-Lipschitz Weisfeiler–Leman (WL) separation power.

Future research directions include further exploration of the FSW embedding as a building block for learning tasks involving non-Euclidean data, as well as extensions to broader notions of distance, such as partial or unbalanced optimal transport.

**Reproducibility Statement**  All experiments in this paper are fully reproducible. The code for training and evaluation, along with the datasets and actual numerical results presented in this paper,

---

[5]These results were obtained using a standard PyTorch implementation of PointNet (Xia, 2019). The original result of PointNet's TensorFlow implementation is 89.2 % accuracy.

are available at the URL https://github.com/tal-amir/FSW-embedding. While we did not use fixed random seeds for our experiments, the results are consistent across multiple runs. Appendix C provides further technical details regarding our experiments.

**Acknowledgements**    The authors are partially funded by ISF grant 272/23. We thank Samantha Chen for her assistance in reproducing the experiments from (Chen & Wang, 2023).

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

The appendices are organized as follows: Appendix A provides additional details on the FSW embedding beyond those covered in the main text. Appendix B provides proofs for our theoretical results. Appendix C presents technical details of our numerical experiments.

# A  FURTHER DETAILS ON THE FSW EMBEDDING

In this appendix, we discuss additional aspects of the FSW embedding. In Appendix A.1, we extend $E_m^{\text{FSW}}$ from probability distributions to measures of arbitrary total mass while retaining injectivity. In Appendix A.2, we present explicit formulas for practical computation and analyze the computational complexity of the embedding.

## A.1  EXTENSION TO MEASURES WITH ARBITRARY TOTAL MASS

Here, we extend the definition of the FSW embedding to measures that are not necessarily probability measures—that is, measures whose weights do not sum to 1—while maintaining injectivity. This extension is crucial for distinguishing multisets of different sizes when their element proportions are identical (as in footnote 3, Page 4), as well as for ensuring continuity when handling the empty multiset as input.

This is particularly important when using the embedding as a message aggregation function in graph neural networks, since (1) in graph-learning tasks, vertex neighborhoods of different sizes but with the same neighbor proportions are typically treated as distinct, and (2) isolated vertices must aggregate messages over the empty multiset. In common graph architectures that require weight normalization, this issue is usually addressed by adding self-loops to ensure that no vertex has an empty neighborhood. The solution presented here eliminates the need for such modifications to the graph structure.

Denote by $\mathcal{M}_{\leq N}(\Omega)$ the collection of all measures with at most $N$ support points over a measurable domain $\Omega \subseteq \mathbb{R}^d$. Any measure $\mu \in \mathcal{M}_{\leq N}(\Omega)$ can be represented by points $\boldsymbol{x}^{(1)}, \dots, \boldsymbol{x}^{(N)} \in \Omega$ and corresponding weights $w_1, \dots, w_N \geq 0$ as a weighted sum of Dirac's delta functions

$$\mu = \sum_{i=1}^{N} w_i \delta_{\boldsymbol{x}^{(i)}}. \tag{14}$$

**Definition.** Let $\mu$ be as in (14). The *total mass* of $\mu$ is defined as

$$\mu(\Omega) = \sum_{i=1}^{N} w_i.$$

A first step towards extending $E^{\text{FSW}}$ from $\mathcal{P}_{\leq N}(\Omega)$ to $\mathcal{M}_{\leq N}(\Omega)$ while maintaining injectivity is to simply add an extra output coordinate that encodes the total mass of the input measure. Define $\hat{E}_m^{\text{FSW}} : \mathcal{M}_{\leq N}(\Omega) \setminus \boldsymbol{0} \to \mathbb{R}^m$ for an input $\mu$ as in (14) by

$$\hat{E}_m^{\text{FSW}}(\mu) := \left[ \mu(\Omega), E_{m-1}^{\text{FSW}}(\hat{\mu}) \right], \tag{15}$$

where $\hat{\mu}$ is the measure $\mu$ normalized to have a total mass of 1:

$$\hat{\mu} := \frac{\mu}{\mu(\Omega)} = \sum_{i=1}^{N} \left( \frac{w_i}{\sum_{j=1}^{N} w_j} \right) \delta_{\boldsymbol{x}^{(i)}}. \tag{16}$$

It is easy to verify that with the above definition, by Theorem 4.1, if $m \geq 2Nd + 2N$ then $\hat{E}_m^{\text{FSW}}$ is injective on $\mathcal{M}_{\leq N}(\mathbb{R}^d) \setminus \boldsymbol{0}$, i.e., the collection of all measures in $\mathcal{M}_{\leq N}(\mathbb{R}^d)$ excluding the zero measure $\mu = \boldsymbol{0}$, for which $\hat{\mu}$ is not defined.[6] Moreover, it further follows from the theorem that $m \geq 2Nd + 2$ suffices to ensure that $\hat{E}_m^{\text{FSW}}$ distinguishes between distinct pairs of input multisets of size $\leq N$, even if their element proportions are identical and they differ only in their cardinality.

---

[6]Strictly speaking, it is injective for *almost any choice* of the embedding parameters $\left( \boldsymbol{v}^{(k)}, \xi^{(k)} \right)_{k=1}^{m-1}$, as in the statement of Theorem 4.1. We ignore this here for brevity.

One limitation of the definition in (15) is that $\hat{E}_m^{\text{FSW}}$ is not defined at the zero measure $\mu = \mathbf{0}$ and, moreover, has a jump discontinuity there. This issue can be remedied by padding input measures whose total mass is below a chosen threshold, assigning the complementary mass to the zero vector in $\mathbb{R}^d$. The result is then normalized to a probability distribution, as before.

Namely, we choose a threshold $\rho > 0$ and define the *regularized measure* $\mu_\rho$ for $\mu \in \mathcal{M}_{\leq N}(\mathbb{R}^d)$ by

$$\mu_\rho := \begin{cases} \frac{\mu}{\mu(\Omega)} & \mu(\Omega) \geq \rho, \\ \left(1 - \frac{\mu(\Omega)}{\rho}\right)\delta_{\mathbf{0}} + \frac{\mu}{\rho} & \mu(\Omega) < \rho, \end{cases} \tag{17}$$

where $\delta_{\mathbf{0}}$ is Dirac's delta function at $0 \in \mathbb{R}^d$. We then redefine $\hat{E}_m^{\text{FSW}}$ to use $\mu_\rho$ rather than $\hat{\mu}$:

$$\hat{E}_m^{\text{FSW}}(\mu) := \left[\mu(\Omega),\ E_{m-1}^{\text{FSW}}(\mu_\rho)\right]. \tag{18}$$

With the definition in (18), $\hat{E}_m^{\text{FSW}}$ is well-defined on the whole of $\mathcal{M}_{\leq N}(\mathbb{R}^d)$. Moreover, it is continuous and has finite directional derivatives everywhere, including the zero measure. As discussed above, if $m \geq 2Nd + 2N$, then $\hat{E}_m^{\text{FSW}}$ is injective, and for injectivity on multisets, $m \geq 2Nd + 2$ suffices.

Note that $\rho$ does not have to be small. In fact, very small values of $\rho$ lead to large gradients of $\hat{E}_m^{\text{FSW}}$ near the zero measure. In our experiments with a message-passing neural network based on the FSW embedding (Sverdlov et al., 2024), we found that values in the range $0.1 \leq \rho \leq 1$ work well in practice.

**Maintaining homogeneity**   In some settings, it is desirable to have an embedding that is positively homogeneous with respect to the input points[7] while still encoding the input multiset size. This property is particularly useful in architectures designed to approximate multiset functions that are known to be homogeneous, yet required to distinguish multisets of different sizes and accommodate the empty multiset.

Note that this necessitates a slight compromise of the injectivity, as any such embedding inevitably maps all measures supported on the zero vector $0 \in \mathbb{R}^d$ to the zero vector $0 \in \mathbb{R}^m$, regardless of their total mass.

To achieve this homogeneity, a straightforward solution is to use a slight modification of (18) and multiply the first coordinate $\mu(\Omega)$ by the norm of the internal embedding $E_{m-1}^{\text{FSW}}(\mu_\rho)$:

$$\hat{E}_m^{\text{FSW}}(\mu) := \left[\left\|E_{m-1}^{\text{FSW}}(\mu_\rho)\right\| \cdot \mu(\Omega),\ E_{m-1}^{\text{FSW}}(\mu_\rho)\right]. \tag{19}$$

With the formulation (19), the positive homogeneity of $E_{m-1}^{\text{FSW}}$ implies that $\hat{E}_m^{\text{FSW}}$ is also positively homogeneous. Moreover, it can be verified that $\hat{E}_m^{\text{FSW}}$ with this formulation still distinguishes between any pair of distinct input measures, except for the case that both measures are supported at $0 \in \mathbb{R}^d$.

## A.2   PRACTICAL COMPUTATION

Here we present formulas that facilitate the practical computation of $E^{\text{FSW}}$, and then analyze its computational complexity.

We start by introducing notation that shall be used to express quantile functions of distributions in $\mathcal{P}_{\leq N}(\mathbb{R})$.

**Definition A.1.** For a vector $\boldsymbol{x} = (x_1, \dots, x_N) \in \mathbb{R}^N$, the *order statistics* $x_{(1)}, \dots, x_{(N)}$ are the coordinates of $\boldsymbol{x}$ sorted in increasing order: $x_{(1)} \leq \dots \leq x_{(N)}$. We define the sorting permutation

$$\sigma(\boldsymbol{x}) = (\sigma_1(\boldsymbol{x}), \dots, \sigma_N(\boldsymbol{x})) \in S_N$$

to be a permutation that satisfies $x_{\sigma_i(\boldsymbol{x})} = x_{(i)}$ for all $i \in [N]$, with ties broken arbitrarily.

---

[7]See definition in Section 4

We now show how $Q_\mu(t)$ can be expressed explicitly in terms of the order statistics of the support of $\mu$. Let

$$\mu = \sum_{i=1}^{N} w_i x_i \in \mathcal{P}_{\leq N}(\mathbb{R})$$

and denote $\boldsymbol{x} = (x_1, \ldots, x_N)$, $\boldsymbol{w} = (w_1, \ldots, w_N)$. Then for all $t \in [0, 1)$, it can be shown that

$$Q_\mu(t) = x_{(k_{\min}(\sigma(\boldsymbol{x}), \boldsymbol{w}, t))}, \tag{20}$$

where $k_{\min}(\sigma, \boldsymbol{w}, t)$ is defined for $\sigma = (\sigma_1, \ldots, \sigma_N) \in S_N$ by

$$k_{\min}(\sigma, \boldsymbol{w}, t) := \min \{k \in [N] \mid w_{\sigma_1} + \cdots + w_{\sigma_k} > t\}. \tag{21}$$

It can be seen in (20) and (21) that $Q_\mu(t)$ is monotone increasing with respect to $t$. Moreover,

$$Q_\mu(0) = \operatorname{ess\,min}(\mu) \quad \text{and} \quad \lim_{t \nearrow 1} Q_\mu(t) = \operatorname{ess\,max}(\mu),$$

with $\operatorname{ess\,min}(\mu)$ and $\operatorname{ess\,max}(\mu)$ denoting the essential minimum and maximum of the distribution $\mu$. We thus augment the definition of $Q_\mu$ to $[0, 1]$ by setting $Q_\mu(1) = \operatorname{ess\,max}(\mu)$.

**Note.** With the above definition, $Q_\mu(t)$ is right-continuous on $[0, 1]$, is continuous at both end points, and since it is monotone increasing, it only has jump discontinuities.

Using the identity (20), $E(\mu; \boldsymbol{v}, \xi)$ can be expressed as

$$
\begin{aligned}
E(\mu; \boldsymbol{v}, \xi) =& 2(1 + \xi) \sum_{k=1}^{N} \int_{t = \sum_{i=1}^{k-1} w_{\sigma_i(\boldsymbol{v}^T \boldsymbol{X})}}^{\sum_{i=1}^{k} w_{\sigma_i(\boldsymbol{v}^T \boldsymbol{X})}} Q_{\boldsymbol{v}^T \mu}(t) \cos(2\pi \xi t) dt \\
=& 2(1 + \xi) \sum_{k=1}^{N} \int_{t = \sum_{i=1}^{k-1} w_{\sigma_i(\boldsymbol{v}^T \boldsymbol{X})}}^{\sum_{i=1}^{k} w_{\sigma_i(\boldsymbol{v}^T \boldsymbol{X})}} \left(\boldsymbol{v}^T \boldsymbol{X}\right)_{(k)} \cos(2\pi \xi t) dt \\
=& 2 \frac{1 + \xi}{2\pi \xi} \sum_{k=1}^{N} \left(\boldsymbol{v}^T \boldsymbol{X}\right)_{(k)} [\sin(2\pi \xi t)]_{t = \sum_{i=1}^{k-1} w_{\sigma_i(\boldsymbol{v}^T \boldsymbol{X})}}^{\sum_{i=1}^{k} w_{\sigma_i(\boldsymbol{v}^T \boldsymbol{X})}},
\end{aligned}
\tag{22}
$$

under the notion $\sum_{i=1}^{0} w_{\sigma_i(\boldsymbol{v}^T \boldsymbol{X})} = 0$. Rearranging terms gives us the alternative formula

$$E(\mu; \boldsymbol{v}, \xi) = 2 \frac{1 + \xi}{2\pi \xi} \sum_{k=1}^{N} \sin \left(2\pi \xi \sum_{i=1}^{k} w_{\sigma_i(\boldsymbol{v}^T \boldsymbol{X})}\right) \left[\left(\boldsymbol{v}^T \boldsymbol{X}\right)_{(k)} - \left(\boldsymbol{v}^T \boldsymbol{X}\right)_{(k+1)}\right], \tag{23}$$

with the definition of $\left(\boldsymbol{v}^T \boldsymbol{X}\right)_{(k)}$ extended to $k = N + 1$ by

$$\left(\boldsymbol{v}^T \boldsymbol{X}\right)_{(N+1)} := 0.$$

To prevent numerical instability for values of $\xi$ near zero, the explicit division by $2\pi \xi$ can be avoided by replacing the sine function in (23) with the *normalized sinc function*

$$\operatorname{sinc}(x) := \begin{cases} \frac{\sin(\pi x)}{\pi x}, & x \neq 0, \\ 0, & x = 0, \end{cases}$$

which is implemented in a numerically stable way in most standard numerical libraries.

This yields the alternative formulas to (22) and (23):

$$E(\mu; \boldsymbol{v}, \xi) = 2(1 + \xi) \sum_{k=1}^{N} \left(\boldsymbol{v}^T \boldsymbol{X}\right)_{(k)} [t \cdot \operatorname{sinc}(2\xi t)]_{t = \sum_{i=1}^{k-1} w_{\sigma_i(\boldsymbol{v}^T \boldsymbol{X})}}^{\sum_{i=1}^{k} w_{\sigma_i(\boldsymbol{v}^T \boldsymbol{X})}}, \tag{24}$$

and

$$E(\mu; \boldsymbol{v}, \xi) = 2(1 + \xi) \sum_{k=1}^{N} \left(\sum_{i=1}^{k} w_{\sigma_i(\boldsymbol{v}^T \boldsymbol{X})}\right) \operatorname{sinc} \left(2\xi \sum_{i=1}^{k} w_{\sigma_i(\boldsymbol{v}^T \boldsymbol{X})}\right) \left[\left(\boldsymbol{v}^T \boldsymbol{X}\right)_{(k)} - \left(\boldsymbol{v}^T \boldsymbol{X}\right)_{(k+1)}\right]. \tag{25}$$

Using the above formulas, the following theorem specifies the complexity of computing $E_m^{\text{FSW}}\left(\mu; \left(\boldsymbol{v}^{(k)}, \xi^{(k)}\right)_{k=1}^{m}\right)$.

**Theorem A.2.** *The function $E_m^{\text{FSW}}\left(\mu; \left(\boldsymbol{v}^{(k)}, \xi^{(k)}\right)_{k=1}^m\right) : \mathcal{P}_{\leq N}\left(\mathbb{R}^d\right) \to \mathbb{R}^m$ can be computed in $\mathcal{O}(mNd + mN \log N)$ time.*

*Proof.* Computing $E_m^{\text{FSW}}\left(\mu; \left(\boldsymbol{v}^{(k)}, \xi^{(k)}\right)_{k=1}^m\right)$ consists of the computation of $E^{\text{FSW}}\left(\mu; \boldsymbol{v}^{(k)}, \xi^{(k)}\right)$ independently for $k = 1, \ldots, m$. In (23) it can be seen that the most expensive operations in each such computation are: (1) the computation of $N$ inner products in $\mathbb{R}^d$, and (2) finding the sorting permutation of these inner products. Therefore, each such computation takes $\mathcal{O}(Nd + N \log N)$ time. $\qquad\square$

# B    PROOFS

This appendix contains proofs for the theoretical results presented in the main text. Appendix B.2 provides proofs for our probabilistic results Theorem 3.2 and Corollary 3.3, which rely on properties of the cosine transform discussed in Appendix B.1. Readers interested only in injectivity and bi-Lipschitzness may skip directly to Appendix B.3, which contains proofs for the results from Section 4.

## B.1    THE COSINE TRANSFORM

The cosine transform takes a major role in our proofs. We now define it and present some of its properties. The results in this subsection appear in standard textbooks such as (Jones, 2001; Boas, 2006). We include them here for completeness.

In the following discussion, $L^p$ denotes the space $L^p(\mathbb{R})$, defined by

$$L^p(\mathbb{R}) := \{f : \mathbb{R} \to \mathbb{R} \mid f \text{ is Lebesgue measurable and } \|f\|_{L^p} < \infty\},$$

with

$$\|f\|_{L^p} := \begin{cases} \left[\int_{\mathbb{R}} |f(t)|^p dt\right]^{1/p} & p \in [1, \infty) \\ \operatorname{ess\,sup}_{t \in \mathbb{R}} |f(t)| & p = \infty. \end{cases}$$

**Definition B.1.** Let $f \in L^1$ such that $f(t) = 0$ for all $t < 0$. Define the *cosine transform* of $f$ is

$$\hat{f}(\xi) := 2 \int_0^\infty f(t) \cos(2\pi \xi t) dt \tag{26}$$

for $\xi \geq 0$.

Note that if $f \in L^1$, then

$$\left\|\hat{f}\right\|_{L^\infty} \leq 2\|f\|_{L^1} \tag{27}$$

since

$$\left|\hat{f}(\xi)\right| \leq 2 \int_0^\infty |f(t)| \cdot |\cos(2\pi \xi t)| dt \leq 2 \int_0^\infty |f(t)| dt = 2\|f\|_{L^1}. \tag{28}$$

Thus, if $f \in L^1$, then $\hat{f} \in L^\infty$. The following lemma proves a better bound as $\xi \to \infty$ if $f$ is monotonous, and shows that the cosine transform preserves the $L^2$-norm.

**Lemma B.2** (Properties of the cosine transform). *Let $f \in L^1$ such that $f(t) = 0$ for all $t < 0$. Then:*

1. *If $f \in L^1 \cap L^2$ then*

$$\int_0^\infty (f(t))^2 dt = \int_0^\infty \left(\hat{f}(t)\right)^2 dt. \tag{29}$$

2. *Suppose that $f \in L^1 \cap L^\infty$, and that $f$ is monotonous on an interval $I = (0, T)$ and vanishes almost everywhere outside of $I$. Then for any $\xi > 0$,*

$$\left|\hat{f}(\xi)\right| \leq \frac{3}{\pi \xi} \|f\|_{L^\infty}. \tag{30}$$

*Proof.* We start from part 1. Throughout the proof, we consider the natural extension of $\hat{f}(\xi)$ to negative values of $\xi$ according to (26), namely

$$\hat{f}(-\xi) = \hat{f}(\xi).$$

Let $f_e(t)$ be the *even component* of $f$,

$$f_e(t) := \tfrac{1}{2}(f(t) + f(-t)) = \tfrac{1}{2}f(|t|).$$

Then

$$
\begin{aligned}
\widehat{f_e}(\xi) &:= \int_{-\infty}^{\infty} f_e(t) e^{-2\pi i \xi t} dt \overset{\text{(a)}}{=} \int_{-\infty}^{\infty} f_e(t) \cos\left(-2\pi \xi t\right) dt \\
&= \int_{-\infty}^{\infty} \tfrac{1}{2}(f(t) + f(-t)) \cos\left(-2\pi \xi t\right) dt \\
&= \tfrac{1}{2} \int_{-\infty}^{0} (f(t) + f(-t)) \cos\left(-2\pi \xi t\right) dt + \tfrac{1}{2} \int_{0}^{\infty} (f(t) + f(-t)) \cos\left(-2\pi \xi t\right) dt \\
&= \tfrac{1}{2} \int_{-\infty}^{0} f(-t) \cos\left(-2\pi \xi t\right) dt + \tfrac{1}{2} \int_{0}^{\infty} f(t) \cos\left(-2\pi \xi t\right) dt \\
&\overset{r=-t}{=\!=\!=\!=} \tfrac{1}{2} \int_{\infty}^{0} f(r) \cos\left(2\pi \xi r\right)(-dr) + \tfrac{1}{2} \int_{0}^{\infty} f(t) \cos\left(2\pi \xi t\right) dt \\
&= \int_{0}^{\infty} f(t) \cos\left(2\pi \xi t\right) dt = \tfrac{1}{2} \hat{f}(\xi),
\end{aligned}
$$

with (a) holding since the Fourier transform of a real even function is real. Hence,

$$
\hat{f}(\xi) = 2\widehat{f_e}(\xi), \qquad \xi \in \mathbb{R}. \tag{31}
$$

Now,

$$
\begin{aligned}
\int_{0}^{\infty} \left(\hat{f}(\xi)\right)^2 d\xi &= \tfrac{1}{2} \int_{-\infty}^{\infty} \left(\hat{f}(\xi)\right)^2 d\xi \\
&= \tfrac{1}{2} \left\|\hat{f}\right\|_{L^2}^2 \overset{\text{(a)}}{=} 2 \left\|\widehat{f_e}\right\|_{L^2}^2 \overset{\text{(b)}}{=} 2 \|f_e\|_{L^2}^2 \overset{\text{(c)}}{=} \|f\|_{L^2}^2 \\
&= \int_{-\infty}^{\infty} (f(t))^2 dt = \int_{0}^{\infty} (f(t))^2 dt,
\end{aligned}
$$

where (a) is by (31), (b) holds by the Plancherel theorem, and (c) holds since

$$
\begin{aligned}
\|f_e\|_{L^2}^2 &= \int_{-\infty}^{\infty} (f_e(t))^2 dt = \int_{-\infty}^{\infty} \left(\tfrac{1}{2}(f(t) + f(-t))\right)^2 dt \\
&= \int_{-\infty}^{\infty} \left[\tfrac{1}{4}(f(t))^2 + \tfrac{1}{2} f(t) f(-t) + \tfrac{1}{4}(f(-t))^2\right] dt \\
&= \tfrac{1}{4} \int_{-\infty}^{\infty} \left[(f(t))^2 + (f(-t))^2\right] dt \\
&= \tfrac{1}{4} \int_{0}^{\infty} (f(t))^2 dt + \tfrac{1}{4} \int_{-\infty}^{0} (f(-t))^2 dt \\
&= \tfrac{1}{2} \int_{0}^{\infty} (f(t))^2 dt = \tfrac{1}{2} \int_{-\infty}^{\infty} (f(t))^2 dt = \tfrac{1}{2} \|f\|_{L^2}^2,
\end{aligned}
$$

and thus part 1 holds. We now prove part 2.

Suppose first that $f$ is differentiable on $I$. Using integration by parts, we have

$$
\begin{aligned}
\hat{f}(\xi) &= 2 \int_{0}^{T} f(t) \cos\left(2\pi \xi t\right) dt \\
&= \frac{1}{\pi \xi} \cdot \overbrace{\left[f(t) \sin\left(2\pi \xi t\right)\right]_{t=0}^{T}}^{A_1} - \frac{1}{\pi \xi} \cdot \overbrace{\int_{0}^{T} f'(t) \sin\left(2\pi \xi t\right) dt}^{A_2}.
\end{aligned}
$$

Let us now bound the terms $A_1$ and $A_2$.

$$
|A_1| = |f(T) \sin\left(2\pi \xi T\right) - f(0) \cdot 0| \le |f(T)| \le \|f\|_{L^\infty},
$$

and

$$|A_2| = \left| \int_0^T f'(t) \sin(2\pi\xi t) dt \right|$$

$$\leq \int_0^T |f'(t)| \cdot |\sin(2\pi\xi t)| dt$$

$$\leq \int_0^T |f'(t)| dt \overset{(a)}{=} \left| \int_0^T f'(t) dt \right|$$

$$= |f(T) - f(0)| \leq 2\|f\|_{L^\infty},$$

with (a) holding since $f'$ does not change sign on $(0, T)$ due to the monotonicity of $f$.

In conclusion, we have

$$\left| \hat{f}(\xi) \right| \leq \frac{1}{\pi\xi}(|A_1| + |A_2|) \leq \frac{3}{\pi\xi}\|f\|_{L^\infty}. \tag{32}$$

To remove the differentiability assumption on $f$, we use the technique of *mollification*, and replace $f$ by a sequence of smooth functions that converges to $f$ in $L^1$; see Chapter 7, Section C.3 of (Jones, 2001).

For the smooth functions to be monotonous, we first define a modified function $\tilde{f} : \mathbb{R} \to \mathbb{R}$

$$\tilde{f}(t) := \begin{cases} f(0^+) & t \leq 0 \\ f(t) & t \in (0, T) \\ f(T^-) & t \geq T, \end{cases} \tag{33}$$

where $f(0^+)$ and $f(T^-)$ are the one-sided limits of $f(t)$ as $t$ approaches $0, T$ respectively from within the interval $(0, T)$. Note that the monotonicity of $f$, combined with the fact that $f \in L^\infty$, implies that both limits must be finite.

Now, $\tilde{f}$ coincides with $f$ on $I$, is monotonous on $\mathbb{R}$, and by construction,

$$\left\| \tilde{f} \right\|_{L^\infty} = \|f\|_{L^\infty}.$$

Let $\phi_\varepsilon : \mathbb{R} \to \mathbb{R}$ for $\varepsilon > 0$ be the mollifying function defined in (Jones, 2001), page 176. We now list a few properties of $\phi_\varepsilon$.

1. $\phi_\varepsilon$ is infinitely differentiable and compactly supported.

2. $\phi_\varepsilon$ is radial, i.e. $\phi_\varepsilon(t) = \phi_\varepsilon(-t)$.

3. $\phi_\varepsilon(t) \geq 0$ for all $t$, and $\phi_\varepsilon(t) > 0$ iff $|t| < \varepsilon$.

4. $\int_\mathbb{R} \phi_\varepsilon(t) dt = 1$.

Let $f_\varepsilon : \mathbb{R} \to \mathbb{R}$ for $\varepsilon > 0$ be defined by

$$f_\varepsilon(t) := \chi_I(t) \int_\mathbb{R} \tilde{f}(r)\phi_\varepsilon(t-r) dr = \chi_I(t) \int_\mathbb{R} \tilde{f}(t+r)\phi_\varepsilon(r) dr, \tag{34}$$

with $\chi_I$ denoting the characteristic function of $I$. From the rightmost part of (34), it is evident that the monotonicity of $\tilde{f}$ implies that $f_\varepsilon$ is monotonous on $I$.

Also note that

$$|f_\varepsilon(t)| \leq \chi_I(t) \int_\mathbb{R} \left| \tilde{f}(t+r) \right| \phi_\varepsilon(r) dr$$

$$\leq \left\| \tilde{f} \right\|_{L^\infty} \int_\mathbb{R} \phi_\varepsilon(r) dr \tag{35}$$

$$= \left\| \tilde{f} \right\|_{L^\infty} = \|f\|_{L^\infty}.$$

Thus,

$$\|f_\varepsilon\|_{L^1} \le T\|f\|_{L^\infty}, \qquad \|f_\varepsilon\|_{L^\infty} \le \|f\|_{L^\infty}, \tag{36}$$

and hence $f_\varepsilon \in L^1 \cap L^\infty$.

From the discussion in (Jones, 2001), $f_\varepsilon$ satisfies:

1. $f_\varepsilon \in C^\infty(I)$

2. $\lim_{\varepsilon \to 0} \|f_\varepsilon - f\|_{L^1} = 0$

So far we have shown that for any $\varepsilon > 0$, $f_\varepsilon$ is in $L^1 \cap L^\infty$, is monotonous and smooth on $I$, and vanishes outside of $I$. Therefore its cosine transform satisfies

$$\left|\hat{f}_\varepsilon(\xi)\right| \overset{(a)}{\le} \frac{3}{\pi\xi}\|f_\varepsilon\|_{L^\infty} \overset{(b)}{\le} \frac{3}{\pi\xi}\|f\|_{L^\infty}, \tag{37}$$

where (a) is by (32) and (b) is by (36). Thus,

$$\begin{aligned}
\tfrac{1}{2}\left|\hat{f}_\varepsilon(\xi) - \hat{f}(\xi)\right| &= \left|\int_0^T (f_\varepsilon(t) - f(t))\cos(2\pi\xi t)dt\right| \\
&\le \|f_\varepsilon - f\|_{L^1}\|\cos(2\pi\xi t)\|_{L^\infty} \\
&\le \|f_\varepsilon - f\|_{L^1} \xrightarrow[\varepsilon \to 0]{} 0.
\end{aligned}$$

In conclusion,

$$\frac{3}{\pi\xi}\|f\|_{L^\infty} \overset{(37)}{\ge} \left|\hat{f}_\varepsilon(\xi)\right| \xrightarrow[\varepsilon \to 0]{} \left|\hat{f}(\xi)\right|$$

and therefore

$$\left|\hat{f}(\xi)\right| \le \frac{3}{\pi\xi}\|f\|_{L^\infty}.$$

$\square$

## B.2 PROBABILISTIC PROPERTIES

In this subsection, we prove our probabilistic claims on $E^{\mathrm{FSW}}$. First, we show that $E^{\mathrm{FSW}}(\mu; \boldsymbol{v}, \xi)$ is uniformly bounded for all embedding parameters $(\boldsymbol{v}, \xi)$ and input distributions $\mu \in \mathcal{P}_{\le N}(\Omega)$, assuming that $\Omega \subset \mathbb{R}^d$ is bounded. We then proceed to prove Theorem 3.2.

For the discussion below, we extend the definition of quantile functions $Q_\mu : [0, 1) \to \mathbb{R}$ to functions on the whole interval $Q_\mu : \mathbb{R} \to \mathbb{R}$, using the convention $Q_\mu(t) = 0$ for $t \notin [0, 1]$, and defining $Q_\mu(1)$ to be the limit $\lim_{t \nearrow 1} Q_\mu(t)$, which is known to exist, since $Q_\mu$ is bounded and monotonous.

Let us start by defining the notation

$$\Delta(\mu, \tilde{\mu}; \boldsymbol{v}, \xi) := \left|E^{\mathrm{FSW}}(\mu; \boldsymbol{v}, \xi) - E^{\mathrm{FSW}}(\tilde{\mu}; \boldsymbol{v}, \xi)\right|. \tag{38}$$

We define a *pseudonorm* for distributions in $\mathcal{P}_{\le N}(\mathbb{R}^d)$:

$$\|\mu\|_{\mathcal{W}_p} := \mathcal{W}_p(\mu, \delta_0), \quad p \in [1, \infty], \tag{39}$$

where $\delta_0$ denotes Dirac's delta function in $0 \in \mathbb{R}^d$, that is, the distribution that assigns a mass of 1 to the point $0 \in \mathbb{R}^d$. Note that $\|\cdot\|_{\mathcal{W}_p}$ is not a true norm, since $\mathcal{P}_{\le N}(\mathbb{R}^d)$ is not a vector space.

The following claim provides useful bounds on the distance between two distributions and on the quantile function of a projected distribution, in terms of the above pseudonorm.

**Claim B.3.** *For any $\mu, \tilde{\mu} \in \mathcal{P}_{\le N}(\mathbb{R}^d)$ and $\boldsymbol{v} \in \mathbb{S}^{d-1}$, $0 \le t \le 1$,*

$$\mathcal{SW}(\mu, \tilde{\mu}) \le \mathcal{W}(\mu, \tilde{\mu}) \le \|\mu\|_{\mathcal{W}_\infty} + \|\tilde{\mu}\|_{\mathcal{W}_\infty}, \tag{40}$$

$$\left|Q_{\boldsymbol{v}^T\mu}(t)\right| \le \|\mu\|_{\mathcal{W}_\infty}. \tag{41}$$

*Proof.* The left inequality in (40) is a well-known property of the sliced Wasserstein distance; see e.g. Eq. (3.2) of (Bayraktar & Guo, 2021). The right inequality is easy to verify by considering the transport plans that transport each of the distributions $\mu, \tilde{\mu}$ to $\delta_0$ and using the triangle for the Wasserstein distance.

Inequality (41) can be proved by showing that the quantile function $Q_{\boldsymbol{v}^T \mu}$ only admits values that are support points of the projected distribution $\boldsymbol{v}^T \mu$, and any such point $y \in \mathbb{R}$ is an inner product $y = \boldsymbol{v}^T \boldsymbol{x}$ of $\boldsymbol{v}$ with some $\boldsymbol{x} \in \mathbb{R}^d$ that is a support point of $\mu$. $\square$

The following lemma shows that $E^{\text{FSW}}(\mu; \boldsymbol{v}, \xi)$ is indeed bounded, as mentioned above, and lays the groundwork for proving Theorem 3.2.

**Lemma B.4.** *Let* $\mu, \tilde{\mu} \in \mathcal{P}_{\leq N}(\mathbb{R}^d)$ *and* $\boldsymbol{v} \in \mathbb{S}^{d-1}$. *Then*

$$\left| E^{\text{FSW}}(\mu; \boldsymbol{v}, \xi) \right| \leq 3\|\mu\|_{\mathcal{W}_\infty} \qquad \forall \xi \geq 0, \tag{42}$$

$$\mathbb{E}_{\xi \sim \mathcal{D}_\xi}\left[\Delta^2(\mu, \tilde{\mu}; \boldsymbol{v}, \xi)\right] = \mathcal{W}^2(\boldsymbol{v}^T \mu, \boldsymbol{v}^T \tilde{\mu}), \tag{43}$$

$$\text{STD}_{\xi \sim \mathcal{D}_\xi}\left[\Delta^2(\mu, \tilde{\mu}; \boldsymbol{v}, \xi)\right] \leq 3\left(\|\mu\|_{\mathcal{W}_\infty} + \|\tilde{\mu}\|_{\mathcal{W}_\infty}\right)\mathcal{W}(\boldsymbol{v}^T \mu, \boldsymbol{v}^T \tilde{\mu}). \tag{44}$$

*Proof.* By the definitions of $E^{\text{FSW}}$ and the cosine transform in (6), (26) respectively,

$$E^{\text{FSW}}(\mu; \boldsymbol{v}, \xi) = (1 + \xi)\hat{Q}_{\boldsymbol{v}^T \mu}(\xi), \tag{45}$$

where $\hat{Q}_{\boldsymbol{v}^T \mu}$ is the cosine transform of the quantile function $Q_{\boldsymbol{v}^T \mu}$. Since $Q_{\boldsymbol{v}^T \mu}(t)$ is monotonous and vanishes outside of $q \in [0, 1]$, by part 2 of Lemma B.2, we have that

$$\left|\hat{Q}_{\boldsymbol{v}^T \mu}(\xi)\right| \leq \frac{3}{\pi \xi}\|Q_{\boldsymbol{v}^T \mu}\|_{L^\infty} \overset{(41)}{\leq} \frac{3}{\pi \xi}\|\mu\|_{\mathcal{W}_\infty}. \tag{46}$$

By (28),

$$\left|\hat{Q}_{\boldsymbol{v}^T \mu}(\xi)\right| \leq 2\|Q_{\boldsymbol{v}^T \mu}\|_{L^1} \overset{(a)}{\leq} 2\|Q_{\boldsymbol{v}^T \mu}\|_{L^\infty} \leq 2\|\mu\|_{\mathcal{W}_\infty}, \tag{47}$$

with (a) holding since $Q_{\boldsymbol{v}^T \mu}$ is supported on $[0, 1]$. Thus, by (46) and (47),

$$\left|\hat{Q}_{\boldsymbol{v}^T \mu}(\xi)\right| \leq \min\left\{2, \frac{3}{\pi \xi}\right\}\|\mu\|_{\mathcal{W}_\infty}.$$

Combined with (45), the above implies

$$\left| E^{\text{FSW}}(\mu; \boldsymbol{v}, \xi) \right| \leq (1 + \xi)\min\left\{2, \frac{3}{\pi \xi}\right\}\|\mu\|_{\mathcal{W}_\infty} \overset{(a)}{\leq} \left(2 + \frac{3}{\pi}\right)\|\mu\|_{\mathcal{W}_\infty} \leq 3\|\mu\|_{\mathcal{W}_\infty},$$

where (a) can be verified by simple analysis. Thus, (42) holds. Note that since $E^{\text{FSW}}(\mu; \boldsymbol{v}, \xi)$ is bounded as a function of $\xi$, so is $\Delta^2(\mu, \tilde{\mu}; \boldsymbol{v}, \xi)$, and therefore both have finite moments of all orders with respect to $\xi$.

Now,

$$\begin{aligned}
\mathbb{E}_\xi\left[\Delta^2(\mu, \tilde{\mu}; \boldsymbol{v}, \xi)\right] &= \mathbb{E}_\xi\left[\left(E^{\text{FSW}}(\mu; \boldsymbol{v}, \xi) - E^{\text{FSW}}(\tilde{\mu}; \boldsymbol{v}, \xi)\right)^2\right] \\
&= \int_0^\infty \frac{1}{(1 + \xi)^2}\left((1 + \xi)^2\left(\hat{Q}_{\boldsymbol{v}^T \mu}(\xi) - \hat{Q}_{\boldsymbol{v}^T \tilde{\mu}}(\xi)\right)^2\right)d\xi \\
&= \int_0^\infty \left(\hat{Q}_{\boldsymbol{v}^T \mu}(\xi) - \hat{Q}_{\boldsymbol{v}^T \tilde{\mu}}(\xi)\right)^2 d\xi \\
&\overset{(a)}{=} \int_0^\infty \left(Q_{\boldsymbol{v}^T \mu}(t) - Q_{\boldsymbol{v}^T \tilde{\mu}}(t)\right)^2 dt \\
&= \int_0^1 \left(Q_{\boldsymbol{v}^T \mu}(t) - Q_{\boldsymbol{v}^T \tilde{\mu}}(t)\right)^2 dt \\
&\overset{(b)}{=} \mathcal{W}^2(\boldsymbol{v}^T \mu, \boldsymbol{v}^T \tilde{\mu}),
\end{aligned}$$

with (a) following from part 1 of Lemma B.2 and the linearity of the cosine transform, and (b) holding by the identity (4). Thus, (43) holds.

To bound the variance of $\Delta^2(\mu, \tilde{\mu}; \boldsymbol{v}, \xi)$, note that

$$
\begin{aligned}
\operatorname{Var}_{\xi}\left[\Delta^2(\mu, \tilde{\mu}; \boldsymbol{v}, \xi)\right] &= \mathbb{E}_{\xi}\left[\left(\Delta^2(\mu, \tilde{\mu}; \boldsymbol{v}, \xi)\right)^2\right] - \left(\mathbb{E}_{\xi}\left[\Delta^2(\mu, \tilde{\mu}; \boldsymbol{v}, \xi)\right]\right)^2 \\
&\stackrel{(43)}{=} \mathbb{E}_{\xi}\left[\Delta^4(\mu, \tilde{\mu}; \boldsymbol{v}, \xi)\right] - \left(\mathcal{W}^2\left(\boldsymbol{v}^T\mu, \boldsymbol{v}^T\tilde{\mu}\right)\right)^2 \\
&= \mathbb{E}_{\xi}\left[\left(E^{\mathrm{FSW}}(\mu; \boldsymbol{v}, \xi) - E^{\mathrm{FSW}}(\tilde{\mu}; \boldsymbol{v}, \xi)\right)^2 \cdot \Delta^2(\mu, \tilde{\mu}; \boldsymbol{v}, \xi)\right] - \mathcal{W}^4\left(\boldsymbol{v}^T\mu, \boldsymbol{v}^T\tilde{\mu}\right) \\
&\leq \mathbb{E}_{\xi}\left[\left(\left|E^{\mathrm{FSW}}(\mu; \boldsymbol{v}, \xi)\right| + \left|E^{\mathrm{FSW}}(\tilde{\mu}; \boldsymbol{v}, \xi)\right|\right)^2 \cdot \Delta^2(\mu, \tilde{\mu}; \boldsymbol{v}, \xi)\right] - \mathcal{W}^4\left(\boldsymbol{v}^T\mu, \boldsymbol{v}^T\tilde{\mu}\right) \\
&\stackrel{(42)}{\leq} \mathbb{E}_{\xi}\left[\left(3\|\mu\|_{\mathcal{W}_{\infty}} + 3\|\tilde{\mu}\|_{\mathcal{W}_{\infty}}\right)^2 \cdot \Delta^2(\mu, \tilde{\mu}; \boldsymbol{v}, \xi)\right] - \mathcal{W}^4\left(\boldsymbol{v}^T\mu, \boldsymbol{v}^T\tilde{\mu}\right) \\
&= 9\left(\|\mu\|_{\mathcal{W}_{\infty}} + \|\tilde{\mu}\|_{\mathcal{W}_{\infty}}\right)^2 \cdot \mathbb{E}_{\xi}\left[\Delta^2(\mu, \tilde{\mu}; \boldsymbol{v}, \xi)\right] - \mathcal{W}^4\left(\boldsymbol{v}^T\mu, \boldsymbol{v}^T\tilde{\mu}\right) \\
&\stackrel{(43)}{=} 9\left(\|\mu\|_{\mathcal{W}_{\infty}} + \|\tilde{\mu}\|_{\mathcal{W}_{\infty}}\right)^2 \cdot \mathcal{W}^2\left(\boldsymbol{v}^T\mu, \boldsymbol{v}^T\tilde{\mu}\right) - \mathcal{W}^4\left(\boldsymbol{v}^T\mu, \boldsymbol{v}^T\tilde{\mu}\right) \\
&\leq 9\left(\|\mu\|_{\mathcal{W}_{\infty}} + \|\tilde{\mu}\|_{\mathcal{W}_{\infty}}\right)^2 \cdot \mathcal{W}^2\left(\boldsymbol{v}^T\mu, \boldsymbol{v}^T\tilde{\mu}\right),
\end{aligned}
$$

and thus (44) holds.

This concludes the proof of Lemma B.4. $\qquad\square$

Let us now prove Theorem 3.2.

**Theorem 3.2.** (Statement in Section 3) *Let $\mu, \tilde{\mu} \in \mathcal{P}_{\leq N}(B_R)$, with $B_R \subset \mathbb{R}^d$ being the closed $\ell_2$-norm ball of radius $R$, centered at zero. Let $\boldsymbol{v} \sim \operatorname{Uniform}(\mathbb{S}^{d-1})$, $\xi \sim \mathcal{D}_{\xi}$. Then*

$$
\mathbb{E}_{\boldsymbol{v}, \xi}\left[\left|E^{\mathrm{FSW}}(\mu; \boldsymbol{v}, \xi) - E^{\mathrm{FSW}}(\tilde{\mu}; \boldsymbol{v}, \xi)\right|^2\right] = \mathcal{SW}^2(\mu, \tilde{\mu}), \tag{7}
$$

$$
\operatorname{STD}_{\boldsymbol{v}, \xi}\left[\left|E^{\mathrm{FSW}}(\mu; \boldsymbol{v}, \xi) - E^{\mathrm{FSW}}(\tilde{\mu}; \boldsymbol{v}, \xi)\right|^2\right] \leq 13 \cdot R^2. \tag{8}
$$

*Proof.* Eq. (7) holds since

$$
\begin{aligned}
\mathbb{E}_{\boldsymbol{v}, \xi}\left[\Delta^2(\mu, \tilde{\mu}; \boldsymbol{v}, \xi)\right] &= \mathbb{E}_{\boldsymbol{v}}\left[\mathbb{E}_{\xi|\boldsymbol{v}}\left[\Delta^2(\mu, \tilde{\mu}; \boldsymbol{v}, \xi)\right]\right] \\
&\stackrel{(43)}{=} \mathbb{E}_{\boldsymbol{v}}\left[\mathcal{W}^2\left(\boldsymbol{v}^T\mu, \boldsymbol{v}^T\tilde{\mu}\right)\right] \\
&\stackrel{(5)}{=} \mathcal{SW}^2(\mu, \tilde{\mu}).
\end{aligned}
$$

We now prove (8).

$$
\begin{aligned}
\operatorname{Var}_{\boldsymbol{v}, \xi}\left[\Delta^2(\mu, \tilde{\mu}; \boldsymbol{v}, \xi)\right] &\stackrel{(a)}{=} \mathbb{E}_{\boldsymbol{v}}\left[\operatorname{Var}_{\xi|\boldsymbol{v}}\left[\Delta^2(\mu, \tilde{\mu}; \boldsymbol{v}, \xi)\right]\right] + \operatorname{Var}_{\boldsymbol{v}}\left[\mathbb{E}_{\xi|\boldsymbol{v}}\left[\Delta^2(\mu, \tilde{\mu}; \boldsymbol{v}, \xi)\right]\right] \\
&\stackrel{(43)}{=} \mathbb{E}_{\boldsymbol{v}}\left[\operatorname{Var}_{\xi|\boldsymbol{v}}\left[\Delta^2(\mu, \tilde{\mu}; \boldsymbol{v}, \xi)\right]\right] + \operatorname{Var}_{\boldsymbol{v}}\left[\mathcal{W}^2\left(\boldsymbol{v}^T\mu, \boldsymbol{v}^T\tilde{\mu}\right)\right] \\
&\stackrel{(44)}{\leq} \mathbb{E}_{\boldsymbol{v}}\left[9\left(\|\mu\|_{\mathcal{W}_{\infty}} + \|\tilde{\mu}\|_{\mathcal{W}_{\infty}}\right)^2 \cdot \mathcal{W}^2\left(\boldsymbol{v}^T\mu, \boldsymbol{v}^T\tilde{\mu}\right)\right] + \operatorname{Var}_{\boldsymbol{v}}\left[\mathcal{W}^2\left(\boldsymbol{v}^T\mu, \boldsymbol{v}^T\tilde{\mu}\right)\right] \\
&= 9\left(\|\mu\|_{\mathcal{W}_{\infty}} + \|\tilde{\mu}\|_{\mathcal{W}_{\infty}}\right)^2 \cdot \mathbb{E}_{\boldsymbol{v}}\left[\mathcal{W}^2\left(\boldsymbol{v}^T\mu, \boldsymbol{v}^T\tilde{\mu}\right)\right] + \operatorname{Var}_{\boldsymbol{v}}\left[\mathcal{W}^2\left(\boldsymbol{v}^T\mu, \boldsymbol{v}^T\tilde{\mu}\right)\right] \\
&\stackrel{(5)}{=} 9\left(\|\mu\|_{\mathcal{W}_{\infty}} + \|\tilde{\mu}\|_{\mathcal{W}_{\infty}}\right)^2 \cdot \mathcal{SW}^2(\mu, \tilde{\mu}) + \operatorname{Var}_{\boldsymbol{v}}\left[\mathcal{W}^2\left(\boldsymbol{v}^T\mu, \boldsymbol{v}^T\tilde{\mu}\right)\right] \\
&\leq 9\left(\|\mu\|_{\mathcal{W}_{\infty}} + \|\tilde{\mu}\|_{\mathcal{W}_{\infty}}\right)^2 \cdot \mathcal{SW}^2(\mu, \tilde{\mu}) + \mathbb{E}_{\boldsymbol{v}}\left[\mathcal{W}^4\left(\boldsymbol{v}^T\mu, \boldsymbol{v}^T\tilde{\mu}\right)\right] \\
&\stackrel{(40)}{\leq} 9\left(\|\mu\|_{\mathcal{W}_{\infty}} + \|\tilde{\mu}\|_{\mathcal{W}_{\infty}}\right)^2 \cdot \left(\|\mu\|_{\mathcal{W}_{\infty}} + \|\tilde{\mu}\|_{\mathcal{W}_{\infty}}\right)^2 + \mathbb{E}_{\boldsymbol{v}}\left[\left(\|\mu\|_{\mathcal{W}_{\infty}} + \|\tilde{\mu}\|_{\mathcal{W}_{\infty}}\right)^4\right] \\
&= 10\left(\|\mu\|_{\mathcal{W}_{\infty}} + \|\tilde{\mu}\|_{\mathcal{W}_{\infty}}\right)^4,
\end{aligned}
$$

where (a) is by (Wasserman, 2004, Theorem 3.27, pg. 55). Now, since

$$
\|\mu\|_{\mathcal{W}_{\infty}}, \|\tilde{\mu}\|_{\mathcal{W}_{\infty}} \leq R,
$$

we have

$$\mathrm{Var}_{\boldsymbol{v},\xi}\big[\Delta^2(\mu,\tilde{\mu};\boldsymbol{v},\xi)\big] \leq 10\big(\|\mu\|_{\mathcal{W}_\infty} + \|\tilde{\mu}\|_{\mathcal{W}_\infty}\big)^4$$
$$\leq 10 \cdot (2R)^4 = 160 \cdot R^4 \leq 13^2 \cdot R^4$$

and therefore (8) holds. □

### B.3 INJECTIVITY AND BI-LIPSCHITZNESS

**Theorem 4.1.** (Statement in Section 4) *Let $E_m^{\mathrm{FSW}} : \mathcal{P}_{\leq N}\big(\mathbb{R}^d\big) \to \mathbb{R}^m$ be as in (9), with $\big(\boldsymbol{v}^{(k)}, \xi^{(k)}\big)_{k=1}^m$ drawn i.i.d. from $\mathrm{Uniform}\big(\mathbb{S}^{d-1}\big) \times \mathcal{D}_\xi$. Then:*

1. *If $m \geq 2Nd + 1$, then with probability 1, $E_m^{\mathrm{FSW}}$ is injective on $\mathcal{S}_{\leq N}\big(\mathbb{R}^d\big)$.*

2. *If $m \geq 2Nd + 2N - 1$, then with probability 1, $E_m^{\mathrm{FSW}}$ is injective on $\mathcal{P}_{\leq N}\big(\mathbb{R}^d\big)$.*

*Proof.* Our proof relies on the theory of $\sigma$-subanalytic functions, introduced in (Amir et al., 2023). This theory defines and studies a class of sets called *$\sigma$-subanalytic sets* and a corresponding class of functions called *$\sigma$-subanalytic functions*. These definitions are technically involved and require an elaborate construction, which we do not include here. However, our proof is self-contained, and we state the relevant properties of these constructs. For a more detailed exposition, we refer the interested reader to (Amir et al., 2023, Appendix A).

The main result of this theory, which plays a central role in our proof, is the *Finite Witness Theorem* (Amir et al., 2023, Theorem A.2). This theorem is a powerful tool for proving the injectivity of parametric functions. Essentially, it enables reducing an infinite family of equality constraints to a finite subfamily chosen randomly, while maintaining equivalence with probability 1.

We begin by stating the full version of the Finite Witness Theorem. The new concepts introduced in the theorem statement will be explained below.

**Theorem B.5** (Finite Witness Theorem). *Let $\mathbb{M} \subseteq \mathbb{R}^p$ and $\mathbb{W} \subseteq \mathbb{R}^q$ be $\sigma$-subanalytic sets of dimensions $D$ and $D_{\boldsymbol{\theta}}$ respectively. Let $F : \mathbb{M} \times \mathbb{W} \to \mathbb{R}$ be a $\sigma$-subanalytic function. Define the set*

$$\mathcal{N} := \{\boldsymbol{z} \in \mathbb{M} \mid F(\boldsymbol{z};\boldsymbol{\theta}) = 0, \ \forall \boldsymbol{\theta} \in \mathbb{W}\}.$$

*Suppose that for all $\boldsymbol{z} \in \mathbb{M} \setminus \mathcal{N}$,*

$$\dim\{\boldsymbol{\theta} \in \mathbb{W} \mid F(\boldsymbol{z};\boldsymbol{\theta}) = 0\} \leq D_{\boldsymbol{\theta}} - 1. \tag{48}$$

*Then for generic $\big(\boldsymbol{\theta}^{(1)}, \ldots, \boldsymbol{\theta}^{(D+1)}\big) \in \mathbb{W}^{D+1}$,*

$$\mathcal{N} = \Big\{\boldsymbol{z} \in \mathbb{M} \ \Big| \ F\big(\boldsymbol{z};\boldsymbol{\theta}^{(i)}\big) = 0, \ \forall i = 1, \ldots, D+1\Big\}.$$

*Moreover, if $\mathbb{W}$ is an open and connected subset of $\mathbb{R}^q$, and $F(\boldsymbol{z};\boldsymbol{\theta})$ is analytic as a function of $\boldsymbol{\theta}$ for all fixed $\boldsymbol{z} \in \mathbb{M}$, then condition (48) is not required, as it is automatically satisfied.*

The function $F(\boldsymbol{z};\boldsymbol{\theta})$ in the theorem can be regarded as representing a parametric family of constraints on $\boldsymbol{z}$, parametrized by $\boldsymbol{\theta}$: a point $\boldsymbol{z} \in \mathbb{M}$ belongs to the feasible set $\mathcal{N}$ if it satisfies the constraints $F(\boldsymbol{z};\boldsymbol{\theta}) = 0$ for all parameter vectors $\boldsymbol{\theta} \in \mathbb{W}$. The vectors $\boldsymbol{\theta}$ are called *witness vectors*.

The theorem essentially states that, if one draws a sufficiently large number of witness vectors from $\mathbb{W}$ at random, then with probability 1, that set of vectors is sufficient to fully determine $\mathcal{N}$.

The number of required vectors depends on the dimension of $\mathbb{M}$. The notion of dimension used in the theorem is the *Hausdorff dimension*, which coincides with the standard notion of dimension for vector spaces and smooth manifolds. The condition (48) is called the *dimension deficiency condition*.

Our next step is to show *how* Theorem 4.1 follows from Theorem B.5, assuming the latter's assumptions are satisfied. We then establish that these assumptions indeed hold. First, we prove part 2 of Theorem 4.1. To this end, we must choose the appropriate $\mathbb{M}, \mathbb{W}$ and $F$. First, set

$$\mathbb{M} = \mathbb{R}^{d \times N} \times \mathbb{R}^{d \times N} \times \Delta^N \times \Delta^N,$$

where $\Delta^N$ is the probability simplex in $\mathbb{R}^N$. Namely, $\mathbb{M}$ is the set of parameters required to describe *pairs* of distributions $\mu, \tilde{\mu} \in \mathcal{P}_{\leq N}(\mathbb{R}^d)$:

$$\mathbb{M} = \left\{ \left( \boldsymbol{X}, \tilde{\boldsymbol{X}}, \boldsymbol{w}, \tilde{\boldsymbol{w}} \right) \ \Big| \ \boldsymbol{X}, \tilde{\boldsymbol{X}} \in \mathbb{R}^{d \times N}, \ \boldsymbol{w}, \tilde{\boldsymbol{w}} \in \Delta^N \right\}.$$

Throughout our proof, we use the convention

$$\mu = \sum_{i=1}^{N} w_i \delta_{\boldsymbol{x}^{(i)}}, \quad \tilde{\mu} = \sum_{i=1}^{N} \tilde{w}_i \delta_{\tilde{\boldsymbol{x}}^{(i)}}.$$

Since $\dim \Delta^N = N - 1$, the dimension of $\mathbb{M}$ is

$$\dim \mathbb{M} = 2Nd + 2N - 2. \tag{49}$$

Second, set $\mathbb{W}$ to be the space of embedding parameters

$$\mathbb{W} = \left\{ (\boldsymbol{v}, \xi) \ \big| \ \boldsymbol{v} \in \mathbb{S}^{d-1}, \xi \in (0, \infty) \right\}.$$

Now, define the function $F : \mathbb{M} \times \mathbb{W} \to \mathbb{R}$,

$$F\left( \boldsymbol{X}, \tilde{\boldsymbol{X}}, \boldsymbol{w}, \tilde{\boldsymbol{w}}; \boldsymbol{v}, \xi \right) := E^{\text{FSW}}(\boldsymbol{X}, \boldsymbol{w}; \boldsymbol{v}, \xi) - E^{\text{FSW}}\left( \tilde{\boldsymbol{X}}, \tilde{\boldsymbol{w}}; \boldsymbol{v}, \xi \right), \tag{50}$$

where, by abuse of notation, $E^{\text{FSW}}(\boldsymbol{X}, \boldsymbol{w}; \boldsymbol{v}, \xi)$ denotes $E^{\text{FSW}}(\mu; \boldsymbol{v}, \xi)$ and $E^{\text{FSW}}\left( \tilde{\boldsymbol{X}}, \tilde{\boldsymbol{w}}; \boldsymbol{v}, \xi \right)$ denotes $E^{\text{FSW}}(\tilde{\mu}; \boldsymbol{v}, \xi)$, with $\mu, \tilde{\mu}$ being the measures defined by $(\boldsymbol{X}, \boldsymbol{w})$ and $(\tilde{\boldsymbol{X}}, \tilde{\boldsymbol{w}})$, respectively.

Essentially, $F\left( \boldsymbol{X}, \tilde{\boldsymbol{X}}, \boldsymbol{w}, \tilde{\boldsymbol{w}}; \boldsymbol{v}, \xi \right)$ compares the two input distributions $\mu, \tilde{\mu}$ by considering their one-sample embeddings $E^{\text{FSW}}(\cdot; \boldsymbol{v}, \xi)$ with the parameters $\boldsymbol{v}, \xi$. Specifically, $F\left( \boldsymbol{X}, \tilde{\boldsymbol{X}}, \boldsymbol{w}, \tilde{\boldsymbol{w}}; \boldsymbol{v}, \xi \right)$ equals zero if and only if both distributions $\mu$ and $\tilde{\mu}$ "appear identical" to $E^{\text{FSW}}$ with the embedding parameters $\boldsymbol{v}, \xi$.

The full embedding $E_m^{\text{FSW}}\left( \cdot; \left( \boldsymbol{v}^{(k)}, \xi^{(k)} \right)_{k=1}^m \right)$ uses $m$ pairs of parameters $\left( \boldsymbol{v}^{(k)}, \xi^{(k)} \right)_{k=1}^m$ drawn independently at random. Recall that part 2 of the theorem, which we wish to prove, assumes that

$$m \geq 2Nd + 2N - 1.$$

Combined with (49), the above implies that

$$m \geq \dim \mathbb{M} + 1 = D + 1,$$

and thus $m$ is a sufficient number of witness vectors as required by Theorem B.5. Therefore, provided that the rest of the assumptions of Theorem B.5 are satisfied, the theorem states that, with probability 1 on the embedding parameters $\left( \boldsymbol{v}^{(k)}, \xi^{(k)} \right)_{k=1}^m$, the following holds: for any pair of input distributions $\mu, \tilde{\mu} \in \mathcal{P}_{\leq N}(\mathbb{R}^d)$, if $E_m^{\text{FSW}}\left( \cdot; \left( \boldsymbol{v}^{(k)}, \xi^{(k)} \right)_{k=1}^m \right)$ does not distinguish between $\mu$ and $\tilde{\mu}$, then

$$E^{\text{FSW}}(\mu; \boldsymbol{v}, \xi) = E^{\text{FSW}}(\tilde{\mu}; \boldsymbol{v}, \xi)$$

for *all* parameters $(\boldsymbol{v}, \xi) \in \mathbb{W}$. This, combined with (7), implies that $\mathcal{SW}(\mu, \tilde{\mu}) = 0$, and therefore $\mu = \tilde{\mu}$. Hence, with probability 1, $E_m^{\text{FSW}}$ is injective on $\mathcal{P}_{\leq N}(\mathbb{R}^d)$.

To prove part 1 of the theorem, namely, injectivity on $\mathcal{S}_{\leq N}(\mathbb{R}^d)$, we restrict $\mathbb{M}$ to parameters that describe pairs of multisets in $\mathcal{S}_{\leq N}(\mathbb{R}^d)$. That is,

$$\mathbb{M} = \left\{ \left( \boldsymbol{X}, \tilde{\boldsymbol{X}}, \boldsymbol{w}, \tilde{\boldsymbol{w}} \right) \ \Big| \ \boldsymbol{X}, \tilde{\boldsymbol{X}} \in \mathbb{R}^{d \times N}, \ \boldsymbol{w}, \tilde{\boldsymbol{w}} \in \left\{ \boldsymbol{w}^{(1)}, \ldots, \boldsymbol{w}^{(N)} \right\} \right\},$$

where each $\boldsymbol{w}^{(n)}$ for $n \in [N]$ is a uniform weight vector corresponding to a multiset of size $n$, as defined in (13). Note that with this definition, $\mathbb{M}$ is a finite union of affine spaces of dimension $2Nd$, and thus

$$\dim \mathbb{M} = 2Nd.$$

Recall that part 1 assumes that

$$m \geq 2Nd + 1 = \dim \mathbb{M} + 1,$$

and the rest of the proof follows as for part 2.

It is now left for us to show that $F$ indeed satisfies the remaining assumptions of Theorem B.5. We begin by stating the formal definition of $\sigma$-subanalytic functions.

**Definition B.6.** A function $f : A \to B$, with $A \subseteq \mathbb{R}^m$, $B \subseteq \mathbb{R}^n$, is said to be $\sigma$-*subanalytic*, if $A$ and $B$ are $\sigma$-subanalytic sets, and the *graph of $f$*, defined by

$$\mathrm{Graph}(f) := \{(x, f(x)) \mid x \in A\} \subseteq A \times B$$

is a $\sigma$-subanalytic subset of $\mathbb{R}^m \times \mathbb{R}^n$.

The following proposition presents several properties of $\sigma$-subanalytic sets and functions that will be used in our proof. These results are established in (Amir et al., 2023, Appendix A), with each property accompanied by a reference to its corresponding statement in that paper.

**Proposition B.7** (Properties of $\sigma$-subanalytic sets and functions)**.**

1. *Finite unions, intersections, and Cartesian products of $\sigma$-subanalytic sets are $\sigma$-subanalytic (Proposition A.11).*

2. *Finite sums, products, compositions and concatenations of $\sigma$-subanalytic functions are $\sigma$-subanalytic (Proposition A.15).*

3. *Semialgebraic functions, and in particular, piecewise-linear functions, are $\sigma$-subanalytic (see Figure 3 therein).*

4. *Analytic functions defined on open domains are $\sigma$-subanalytic (Lemma A.14).*

5. *Any $\sigma$-subanalytic set $A$ can be presented as a countable union*

$$A = \bigcup_{i=1}^{\infty} C_i,$$

   *where each $C_i$ is a $C^\infty$ manifold. In particular,*

$$\dim A = \max_{i \in \mathbb{N}} \dim C_i$$

   *(Propositions A.11 and A.16).*

An additional property required for our proof is that the preimage of a $\sigma$-subanalytic set under a $\sigma$-subanalytic function is $\sigma$-subanalytic. We now prove this.

**Proposition B.8.** *Let $f : A \to B$ be a $\sigma$-subanalytic function, and let $C \subseteq B$ be a $\sigma$-subanalytic set. Then the set*

$$f^{-1}(C) := \{a \in A \mid f(a) \in C\}$$

*is $\sigma$-subanalytic.*

*Proof.* The set $f^{-1}(C)$ can be presented as

$$f^{-1}(C) = \pi\big(\mathrm{Graph}(f) \cap (A \times C)\big),$$

with $\pi : \mathbb{R}^m \times \mathbb{R}^n \to \mathbb{R}^m$ being the orthogonal projection operator

$$\pi((x, y)) = x.$$

Thus, it follows from Proposition B.7 that $f^{-1}(C)$ is $\sigma$-subanalytic. $\qquad\square$

Now, we argue that $\mathbb{M}$ and $\mathbb{W}$ are $\sigma$-subanalytic sets. The reason is that both sets are semialgebraic, which is easy to verify, and any semialgebraic set is $\sigma$-subanalytic; see (Amir et al., 2023, Figure 3).

Next, we want to show that $F$ is a $\sigma$-subanalytic function. For this, we need to show that $E^{\mathrm{FSW}}(\boldsymbol{X}, \boldsymbol{w}; \boldsymbol{v}, \xi)$ is $\sigma$-subanalytic as a function of $(\boldsymbol{X}, \boldsymbol{w}, \boldsymbol{v}, \xi)$. To see this, note that by (23), $E^{\mathrm{FSW}}(\boldsymbol{X}, \boldsymbol{w}; \boldsymbol{v}, \xi)$ is the sum of terms of the form

$$2\frac{1+\xi}{2\pi\xi} \sin\left(2\pi\xi \sum_{i=1}^{k} w_{\sigma_i(\boldsymbol{v}^T\boldsymbol{X})}\right) \left[\left(\boldsymbol{v}^T\boldsymbol{X}\right)_{(k)} - \left(\boldsymbol{v}^T\boldsymbol{X}\right)_{(k+1)}\right], \tag{51}$$

with the sum taken over $k \in [N]$. The product $\boldsymbol{v}^T \boldsymbol{X}$ is semialgebraic, and thus is $\sigma$-subanalytic. Each term $\left[ \left( \boldsymbol{v}^T \boldsymbol{X} \right)_{(k)} - \left( \boldsymbol{v}^T \boldsymbol{X} \right)_{(k+1)} \right]$ is piecewise linear in $\boldsymbol{v}^T \boldsymbol{X}$ and thus $\sigma$-subanalytic. Each $\sum_{i=1}^{k} w_{\sigma_i(\boldsymbol{v}^T \boldsymbol{X})}$ is semialgebraic and thus is also $\sigma$-analytic. The product $2\pi\xi \sum_{i=1}^{k} w_{\sigma_i(\boldsymbol{v}^T \boldsymbol{X})}$, composition with the analytic sine function $\sin\left( 2\pi\xi \sum_{i=1}^{k} w_{\sigma_i(\boldsymbol{v}^T \boldsymbol{X})} \right)$, and again product $2\frac{1+\xi}{2\pi\xi} \sin\left( 2\pi\xi \sum_{i=1}^{k} w_{\sigma_i(\boldsymbol{v}^T \boldsymbol{X})} \right)$ is also $\sigma$-subanalytic. Finally, the product (51) and the finite sum of such over $k \in [N]$ is $\sigma$-subanalytic. Therefore, $F$ is $\sigma$-subanalytic, as it the difference of two $\sigma$-subanalytic functions.

Since $F$ is not generally analytic as a function of $(\boldsymbol{v}, \xi)$ for fixed $\left( \boldsymbol{X}, \tilde{\boldsymbol{X}}, \boldsymbol{w}, \tilde{\boldsymbol{w}} \right)$, we need to explicitly show that $F$ satisfies the dimension deficiency condition. Let $\mu, \tilde{\mu} \in \mathcal{P}_{\leq N}\left( \mathbb{R}^d \right)$ be two fixed distributions and suppose that $\mu \neq \tilde{\mu}$. Let $A \subseteq \mathbb{W}$ be the set

$$A := \left\{ (\boldsymbol{v}, \xi) \in \mathbb{S}^{d-1} \times (0, \infty) \ \middle| \ E^{\mathrm{FSW}}(\mu; \boldsymbol{v}, \xi) = E^{\mathrm{FSW}}(\tilde{\mu}; \boldsymbol{v}, \xi) \right\}.$$

We need to show that $A$ is dimension deficient; that is, $\dim A < \dim \mathbb{W}$.

It is sufficient to show that $A$ is a set of measure zero in $\mathbb{W}$. By *measure*, we mean that we consider $\mathbb{W} = \mathbb{S}^{d-1} \times (0, \infty)$ endowed with the product measure, where $\mathbb{S}^{d-1}$ is equipped with the uniform measure and $(0, \infty)$ with the Lebesgue measure.

The reason is as follows: $A$ is $\sigma$-subanalytic since it is the preimage of the $\sigma$-subanalytic set $\{0\}$ under the $\sigma$-subanalytic function $F\left( \boldsymbol{X}, \tilde{\boldsymbol{X}}, \boldsymbol{w}, \tilde{\boldsymbol{w}}; \cdot \right)$. If $\dim A = \dim \mathbb{W}$, then by Proposition B.7, $A$ contains some $C^\infty$ manifold $C_i$ such that $\dim C_i = \dim \mathbb{W}$. Any such manifold is a nonempty subset of $\mathbb{W}$ that is open in the product topology on $\mathbb{S}^{d-1} \times (0, \infty)$. Since $C_i$ is open and nonempty, it has positive measure, and thus so does $A$.

To finish the proof, we now show that $A$ indeed has measure zero in $\mathbb{W}$, namely that almost any $(\boldsymbol{v}, \xi) \in \mathbb{W}$ is not in $A$. First, note that for almost any $\boldsymbol{v} \in \mathbb{S}^{d-1}$, the inner product $\langle \boldsymbol{v}, \cdot \rangle : \mathbb{R}^d \to \mathbb{R}$ maps the finite number of support points of $\mu$ and $\tilde{\mu}$ injectively to $\mathbb{R}$. For all such $\boldsymbol{v}$, the obtained one-dimensional projected measures will be distinct, that is, $\boldsymbol{v}^T \mu \neq \boldsymbol{v}^T \tilde{\mu}$, and therefore, so will be their quantile functions,

$$Q_{\boldsymbol{v}^T \mu} \neq Q_{\boldsymbol{v}^T \tilde{\mu}}.$$

Once $\boldsymbol{v}$ is also fixed, $F\left( \boldsymbol{X}, \tilde{\boldsymbol{X}}, \boldsymbol{w}, \tilde{\boldsymbol{w}}; \boldsymbol{v}, \xi \right)$ is analytic as a function of $\xi$. Thus it must be nonzero almost everywhere. Otherwise, it would be zero at all points $\xi$, which would contradict the fact that $Q_{\boldsymbol{v}^T \mu} \neq Q_{\boldsymbol{v}^T \tilde{\mu}}$ and the cosine transform is an isometry.

Thus, we have shown that $F$ satisfies the dimension deficiency condition. This concludes our proof. $\square$

Let us now prove Theorem 4.4.

**Theorem 4.4.** (Statement in Section 4) *Let* $E : \mathcal{P}_{\leq N}(\Omega) \to \mathbb{R}^m$, *where* $N \geq 2$ *and* $\Omega \subseteq \mathbb{R}^d$ *has a nonempty interior. Then for all* $p \in [1, \infty]$, *$E$ is not bi-Lipschitz on* $\mathcal{P}_{\leq N}(\Omega)$ *with respect to* $\mathcal{W}_p$.

Before proving the theorem, we note that it implies that most practical embeddings of $\mathcal{P}_{\leq N}(\Omega)$ are likely to fail in lower-Lipschitzness, since it is reasonable to expect such embeddings to be upper Lipschitz. This is formulated in the following corollary.

**Corollary B.9.** *Under the assumptions of Theorem 4.4, if* $E : \mathcal{P}_{\leq N}(\Omega) \to \mathbb{R}^m$ *is upper-Lipschitz with respect to* $\mathcal{W}_1$, *then it is not lower-Lipschitz with respect to any* $\mathcal{W}_p$ *with* $p \in [1, \infty]$.

*Proof.* First, note that it is a well-known fact in optimal transport theory (Villani et al., 2008, Remark 6.6) that $\mathcal{W}_p(\mu, \tilde{\mu})$ is monotone increasing in $p$, that is,

$$\mathcal{W}_p(\mu, \tilde{\mu}) \leq \mathcal{W}_q(\mu, \tilde{\mu}), \qquad 1 \leq p \leq q \leq \infty. \tag{52}$$

Now, if $E$ is upper-Lipschitz w.r.t. $\mathcal{W}_1$, then by Theorem 4.4 it is not lower-Lipschitz w.r.t. $\mathcal{W}_1$. Therefore, since $\mathcal{W}_p(\mu, \tilde{\mu}) \geq \mathcal{W}_1(\mu, \tilde{\mu})$ for any $p \geq 1$, $E$ is thus not lower-Lipschitz w.r.t. $\mathcal{W}_p$. $\square$

*Proof of Theorem 4.4.* Our proof consists of three steps. First, in Lemma B.10 below, we prove the theorem for the special case that $E$ is positively homogeneous and $\Omega$ is an open ball centered at zero. Then, in Lemma B.11, we release the homogeneity assumption by considering a homogenized version of $E$. Finally, we generalize to arbitrary $\Omega$ with a nonempty interior in a straightforward manner.

For convenience of notation, before proceeding with the proof, we introduce an alternative scalar multiplication operation for distributions in $\mathcal{P}_{\leq N}(\Omega)$.

**Definition.** For $\mu = \sum_{i=1}^N w_i \delta_{\boldsymbol{x}^{(i)}} \in \mathcal{P}_{\leq N}(\mathbb{R}^d)$ and a scalar $\alpha \in \mathbb{R}$, we define the distribution $\alpha\mu \in \mathcal{P}_{\leq N}(\mathbb{R}^d)$ by

$$\alpha\mu := \sum_{i=1}^N w_i \delta_{\alpha\boldsymbol{x}^{(i)}}.$$

Note that this definition is *not* the same as the standard scalar multiplication operation for measures used in Appendix A.1 (e.g., equation (16)), which scales the measure weights rather than the support points. We will use this alternative notation throughout, and only within, this section.

Let us begin with the special case of a positively homogeneous $E$.

**Lemma B.10.** *Let $E : \mathcal{P}_{\leq N}(\Omega) \to \mathbb{R}^m$, with $\Omega \subseteq \mathbb{R}^d$ being an open ball centered at zero, $N \geq 2$ and $m \geq 1$. Suppose that $E$ is positively homogeneous, i.e. $E(\alpha\mu) = \alpha E(\mu)$ for any $\mu \in \mathcal{P}_{\leq N}(\Omega)$, $\alpha \geq 0$. Then for all $p \in [1, \infty]$, $E$ is not bi-Lipschitz with respect to $\mathcal{W}_p$.*

*Proof.* In our proof we use the pseudonorm $\|\cdot\|_{\mathcal{W}_p}$, defined in (39) for distributions in $\mathcal{P}_{\leq N}(\mathbb{R}^d)$. Note that this pseudonorm is positively homogeneous, namely, for all $\mu \in \mathcal{P}_{\leq N}(\mathbb{R}^d)$, $\alpha \geq 0$ and $p \in [1, \infty]$,

$$\|\alpha\mu\|_{\mathcal{W}_p} = \alpha\|\mu\|_{\mathcal{W}_p}.$$

Let $\{\theta_t\}_{t=1}^\infty$ be a sequence of real numbers such that

$$0 < \theta_{t+1} \leq \tfrac{1}{2}\theta_t \leq 1 \quad \forall t \geq 1. \tag{53}$$

By assumption, the set $\Omega$ contains a ball $B_r(0)$ of radius $r > 0$ centered at zero. Choose $\boldsymbol{x} \neq 0$ in that ball. For $\theta \in [0, 1]$, we define the distribution

$$\mu(\theta) = (1 - \theta)\delta_0 + \theta\delta_{\boldsymbol{x}},$$

where $\delta_0$ is Dirac's delta function at $0 \in \mathbb{R}^d$. That is, $\mu(\theta)$ assigns weights of $1 - \theta$ and $\theta$ to the points $0$ and $\boldsymbol{x}$ in $\mathbb{R}^d$ respectively. It is straightforward to verify that for $1 \leq p < \infty$

$$\mathcal{W}_p(\mu(\theta_t), \delta_0) = [\theta_t\|\boldsymbol{x}\|^p]^{1/p} = \sqrt[p]{\theta_t}\|\boldsymbol{x}\|.$$

This holds for $p = \infty$ too, with the convention $\sqrt[\infty]{x} = 1$ for $x > 0$. Therefore, for all $t \in \mathbb{N}$,

$$\frac{E(\mu(\theta_t)) - E(\delta_0)}{\mathcal{W}_p(\mu(\theta_t), \delta_0)} = \frac{1}{\|\boldsymbol{x}\|} \frac{E(\mu(\theta_t)) - E(\delta_0)}{\sqrt[p]{\theta_t}} = \frac{1}{\|\boldsymbol{x}\|} \frac{E(\mu(\theta_t))}{\sqrt[p]{\theta_t}}, \tag{54}$$

where the last equality follows from the homogeneity of $E$, which implies that $E(\delta_0) = 0$.

We can assume that $E$ in upper-Lipschitz, since otherwise there is nothing to prove. Under this assumption, the norm of the expression in (54) is uniformly upper-bounded for all $t \in \mathbb{N}$, which implies that there exists a subsequence of $\theta_t$ for which this expression converges. Replacing $\theta_t$ with this subsequence, it can be verified that the new subsequence still satisfies (53), and that for an appropriate vector $L \in \mathbb{R}^m$,

$$\lim_{t \to \infty} \frac{E(\mu(\theta_t))}{\sqrt[p]{\theta_t}} = L.$$

Now, consider the sequence of distributions

$$\tilde{\mu}_t := \sqrt[p]{\frac{\theta_t}{\theta_{t-1}}} \cdot \mu(\theta_{t-1}), \quad t \geq 2.$$

Since

$$\sqrt[p]{\frac{\theta_t}{\theta_{t-1}}} \leq \sqrt[p]{\frac{1}{2}} < 1,$$

and $x \in B_r(0) \subseteq \Omega$, the distribution $\tilde{\mu}_t$ indeed belongs to $\mathcal{P}_{\leq N}(\Omega)$. We wish to lower-bound the $p$-Wasserstein distance from $\mu(\theta_t)$ to $\tilde{\mu}_t$ for $t \geq 2$. Note that both distributions divide their mass between zero and an additional vector: the distribution $\tilde{\mu}_t$ assigns a mass of $\theta_{t-1}$ to the nonzero point $\sqrt[p]{\frac{\theta_t}{\theta_{t-1}}} x$, whereas the other distribution $\mu(\theta_t)$ assigns a smaller mass of $\theta_t$ to the nonzero point $x$, with the complementary masses assigned to zero. Therefore, transporting $\tilde{\mu}_t$ to $\mu(\theta_t)$ requires transporting at least a mass of $\theta_{t-1} - \theta_t$ from $\sqrt[p]{\frac{\theta_t}{\theta_{t-1}}} x$ to 0, so that for all $1 \leq p < \infty$

$$\mathcal{W}_p^p(\mu(\theta_t), \tilde{\mu}_t) \geq (\theta_{t-1} - \theta_t) \| \sqrt[p]{\tfrac{\theta_t}{\theta_{t-1}}} x - 0 \|^p$$

$$= \theta_t (1 - \frac{\theta_t}{\theta_{t-1}}) \|x\|^p$$

$$\geq \frac{1}{2} \theta_t \|x\|^p.$$

We obtained that for $p < \infty$,

$$\mathcal{W}_p(\mu(\theta_t), \tilde{\mu}_t) \geq \sqrt[p]{\theta_t / 2} \|x\|, \tag{55}$$

and the same argument can be used to verify that this is the case for $p = \infty$ as well. We conclude that

$$\frac{\|E(\mu(\theta_t)) - E(\tilde{\mu}_t)\|}{\mathcal{W}_p(\mu(\theta_t), \tilde{\mu}_t)} \overset{(a)}{\leq} \frac{\sqrt[p]{\frac{1}{\theta_t}} \left\| E(\mu(\theta_t)) - \sqrt[p]{\frac{\theta_t}{\theta_{t-1}}} E(\mu(\theta_{t-1})) \right\|}{\sqrt[p]{1/2} \|x\|}$$

$$= \frac{\left\| \sqrt[p]{\frac{1}{\theta_t}} E(\mu(\theta_t)) - \sqrt[p]{\frac{1}{\theta_{t-1}}} E(\mu(\theta_{t-1})) \right\|}{\sqrt[p]{1/2} \|x\|} \to 0$$

where (a) is by (55) and the homogeniety of $E$, and the convergence to zero is because both expressions in the numerator converge to the same limit $L$. This shows that $E$ is not lower-Lipschitz, which concludes the proof of Lemma B.10. $\qquad \square$

The following lemma shows that the homogeneity assumption on $E$ can be released.

**Lemma B.11.** *Let $E : \mathcal{P}_{\leq N}(\Omega) \to \mathbb{R}^m$, with $\Omega \subseteq \mathbb{R}^d$ being an open ball centered at zero, $N \geq 2$ and $m \geq 1$. Then for all $p \in [1, \infty]$, $E$ is not bi-Lipschitz with respect to $\mathcal{W}_p$.*

*Proof.* Let $p \in [1, \infty]$ and suppose by contradiction that $E$ is bi-Lipschitz with constants $0 < c \leq C < \infty$,

$$c \cdot \mathcal{W}_p(\mu, \tilde{\mu}) \leq \|E(\mu) - E(\tilde{\mu})\| \leq C \cdot \mathcal{W}_p(\mu, \tilde{\mu}), \qquad \forall \mu, \tilde{\mu} \in \mathcal{P}_{\leq N}(\Omega). \tag{56}$$

We can assume without loss of generality that $E(\delta_0) = 0$, since otherwise, let

$$\tilde{E}(\mu) := E(\mu) - E(\delta_0),$$

then $E$ satisfies (56) if and only if $\tilde{E}$ satisfies (56).

We first prove an auxiliary claim.

**Claim.** *For any $\mu, \tilde{\mu} \in \mathcal{P}_{\leq N}(\Omega)$ with $\|\mu\|_{\mathcal{W}_p} = 1$ and $0 < \|\tilde{\mu}\|_{\mathcal{W}_p} \leq 1$,*

$$\left\| E\left( \frac{\tilde{\mu}}{\|\tilde{\mu}\|_{\mathcal{W}_p}} \right) - E(\tilde{\mu}) \right\| \leq C \cdot \left( 1 - \|\tilde{\mu}\|_{\mathcal{W}_p} \right) \leq C \cdot \mathcal{W}_p(\mu, \tilde{\mu}). \tag{57}$$

*Proof.* By (56),

$$\left\| E\left( \frac{\tilde{\mu}}{\|\tilde{\mu}\|_{\mathcal{W}_p}} \right) - E(\tilde{\mu}) \right\| \leq C \cdot \mathcal{W}_p\left( \frac{\tilde{\mu}}{\|\tilde{\mu}\|_{\mathcal{W}_p}}, \tilde{\mu} \right).$$

We shall now show that

$$\mathcal{W}_p\left( \frac{\tilde{\mu}}{\|\tilde{\mu}\|_{\mathcal{W}_p}}, \tilde{\mu} \right) \leq 1 - \|\tilde{\mu}\|_{\mathcal{W}_p}.$$

Let $\tilde{\mu} = \sum_{i=1}^{N} w_i \delta_{\tilde{\boldsymbol{x}}_i}$ be a parametrization of $\tilde{\mu}$. Consider the transport plan $\pi = (\pi_{ij})_{i,j \in [N]}$ from $\tilde{\mu}$ to $\frac{\tilde{\mu}}{\|\tilde{\mu}\|_{\mathcal{W}_p}}$ given by

$$\pi_{ij} = \begin{cases} w_i & i = j \\ 0 & i \neq j. \end{cases}$$

By definition, $\mathcal{W}_p\left( \frac{\tilde{\mu}}{\|\tilde{\mu}\|_{\mathcal{W}_p}}, \tilde{\mu} \right)$ is smaller or equal to the cost of transporting $\tilde{\mu}$ to $\frac{\tilde{\mu}}{\|\tilde{\mu}\|_{\mathcal{W}_p}}$ according to $\pi$. Thus, for $p < \infty$,

$$\mathcal{W}_p^p\left( \frac{\tilde{\mu}}{\|\tilde{\mu}\|_{\mathcal{W}_p}}, \tilde{\mu} \right) \leq \sum_{i=1}^{N} w_i \left\| \frac{1}{\|\tilde{\mu}\|_{\mathcal{W}_p}} \tilde{\boldsymbol{x}}_i - \tilde{\boldsymbol{x}}_i \right\|^p = \sum_{i=1}^{N} w_i \left\| \left( \frac{1}{\|\tilde{\mu}\|_{\mathcal{W}_p}} - 1 \right) \tilde{\boldsymbol{x}}_i \right\|^p$$

$$= \left( \frac{1}{\|\tilde{\mu}\|_{\mathcal{W}_p}} - 1 \right)^p \sum_{i=1}^{N} w_i \|\tilde{\boldsymbol{x}}_i\|^p = \left( \frac{1}{\|\tilde{\mu}\|_{\mathcal{W}_p}} - 1 \right)^p \|\tilde{\mu}\|_{\mathcal{W}_p}^p$$

$$= \left( 1 - \|\tilde{\mu}\|_{\mathcal{W}_p} \right)^p,$$

and thus

$$\mathcal{W}_p\left( \frac{\tilde{\mu}}{\|\tilde{\mu}\|_{\mathcal{W}_p}}, \tilde{\mu} \right) \leq \left( 1 - \|\tilde{\mu}\|_{\mathcal{W}_p} \right).$$

Both sides of the above inequality are continuous in $p$, including at the limit $p \to \infty$. Thus, the above inequality also holds for $p = \infty$. Now, to show that

$$1 - \|\tilde{\mu}\|_{\mathcal{W}_p} \leq \mathcal{W}_p(\mu, \tilde{\mu}),$$

note that

$$1 - \|\tilde{\mu}\|_{\mathcal{W}_p} = \|\mu\|_{\mathcal{W}_p} - \|\tilde{\mu}\|_{\mathcal{W}_p} = \mathcal{W}_p(\mu, 0) - \mathcal{W}_p(\tilde{\mu}, 0) \leq \mathcal{W}_p(\mu, \tilde{\mu}),$$

where the last inequality is the reverse triangle inequality, since $\mathcal{W}_p(\cdot, \cdot)$ is a metric. Thus, (57) holds. $\square$

Now we define the *homogenized* function $\hat{E} : \mathcal{P}_{\leq N}(\Omega) \to \mathbb{R}^{m+1}$ by

$$\begin{cases} \hat{E}(\mu) := \left[ \|\mu\|_{\mathcal{W}_p}, \|\mu\|_{\mathcal{W}_p} E\left( \frac{\mu}{\|\mu\|_{\mathcal{W}_p}} \right) \right], & \mu \neq \delta_0 \\ 0 & \mu = \delta_0. \end{cases} \tag{58}$$

It is straightforward to verify that $\hat{E}$ is well-defined and positively homogeneous. By Lemma B.10, $\hat{E}$ it is not bi-Lipschitz with respect to $\mathcal{W}_p$, and thus there exist two sequences of distributions $\mu_t, \tilde{\mu}_t \in \mathcal{P}_{\leq N}(\Omega)$, $t \geq 1$, such that

$$\frac{\left\| \hat{E}(\mu_t) - \hat{E}(\tilde{\mu}_t) \right\|}{\mathcal{W}_p(\mu_t, \tilde{\mu}_t)} \xrightarrow[t \to \infty]{} L, \tag{59}$$

with $L = 0$ or $L = \infty$. Since $\hat{E}$ is positively homogeneous, we can assume without loss of generality that

$$1 = \|\mu_t\|_{\mathcal{W}_p} \geq \|\tilde{\mu}_t\|_{\mathcal{W}_p} \quad \text{for all } t \geq 1.$$

This can be seen by dividing each $\mu_t$ and $\tilde{\mu}_t$ by $\max\left\{\|\mu_t\|_{\mathcal{W}_p}, \|\tilde{\mu}_t\|_{\mathcal{W}_p}\right\}$ and swapping $\mu_t$ and $\tilde{\mu}_t$ for all $t$ for which $\|\mu_t\|_{\mathcal{W}_p} < \|\tilde{\mu}_t\|_{\mathcal{W}_p}$.

If $\tilde{\mu}_t = \delta_0$ for an infinite subset of indices $t$, then redefine $\mu_t$ and $\tilde{\mu}_t$ to be the corresponding subsequences with those indices, and now (59) implies that

$$\frac{\|E(\mu_t) - E(\delta_0)\|}{\mathcal{W}_p(\mu_t, \delta_0)} \overset{(a)}{=} \frac{\left\|\hat{E}(\mu_t) - \hat{E}(\delta_0)\right\|}{\mathcal{W}_p(\mu_t, \delta_0)} \xrightarrow[t\to\infty]{} L,$$

with (a) holding since $\|\tilde{\mu}_t\|_{\mathcal{W}_p} = 1$. This contradicts the bi-Lipschitzness of $E$. Thus, we can assume that $\tilde{\mu}_t = \delta_0$ at most at a finite subset of indices $t$. By skipping those indices in $\mu_t$ and $\tilde{\mu}_t$, we can assume without loss of generality that

$$1 = \|\mu_t\|_{\mathcal{W}_p} \geq \|\tilde{\mu}_t\|_{\mathcal{W}_p} > 0 \quad \text{for all } t \geq 1. \tag{60}$$

Let us first handle the case $L = \infty$. The first coordinate of $\hat{E}(\mu_t) - \hat{E}(\tilde{\mu}_t)$ is bounded by

$$\left|\|\mu_t\|_{\mathcal{W}_p} - \|\tilde{\mu}_t\|_{\mathcal{W}_p}\right| = 1 - \|\tilde{\mu}_t\|_{\mathcal{W}_p} \leq \mathcal{W}_p(\mu_t, \tilde{\mu}_t)$$

according to (57). Thus, the violation of bi-Lipschitzness must come from the last $m$ coordinates of $\hat{E}(\mu_t) - \hat{E}(\tilde{\mu}_t)$. That is, (59) with $L = \infty$, combined with the fact that $\|\tilde{\mu}_t\|_{\mathcal{W}_p} > 0 \,\forall t$, implies that

$$\frac{\left\|\|\mu_t\|_{\mathcal{W}_p} E\left(\frac{\mu_t}{\|\mu_t\|_{\mathcal{W}_p}}\right) - \|\tilde{\mu}_t\|_{\mathcal{W}_p} E\left(\frac{\tilde{\mu}_t}{\|\tilde{\mu}_t\|_{\mathcal{W}_p}}\right)\right\|}{\mathcal{W}_p(\mu_t, \tilde{\mu}_t)} \xrightarrow[t\to\infty]{} \infty. \tag{61}$$

On the other hand,

$$\left\|\|\mu_t\|_{\mathcal{W}_p} E\left(\frac{\mu_t}{\|\mu_t\|_{\mathcal{W}_p}}\right) - \|\tilde{\mu}_t\|_{\mathcal{W}_p} E\left(\frac{\tilde{\mu}_t}{\|\tilde{\mu}_t\|_{\mathcal{W}_p}}\right)\right\| \overset{(a)}{=} \left\|E(\mu_t) - \|\tilde{\mu}_t\|_{\mathcal{W}_p} E\left(\frac{\tilde{\mu}_t}{\|\tilde{\mu}_t\|_{\mathcal{W}_p}}\right)\right\|$$

$$\overset{(b)}{\leq} \|E(\mu_t) - E(\tilde{\mu}_t)\| + \left\|E(\tilde{\mu}_t) - E\left(\frac{\tilde{\mu}_t}{\|\tilde{\mu}_t\|_{\mathcal{W}_p}}\right)\right\| + \left\|E\left(\frac{\tilde{\mu}_t}{\|\tilde{\mu}_t\|_{\mathcal{W}_p}}\right) - \|\tilde{\mu}_t\|_{\mathcal{W}_p} E\left(\frac{\tilde{\mu}_t}{\|\tilde{\mu}_t\|_{\mathcal{W}_p}}\right)\right\|, \tag{62}$$

where (a) holds since $\|\mu_t\|_{\mathcal{W}_p} = 1$ and (b) is by the triangle inequality. We shall now upper-bound the three above terms.

First, by (56),

$$\|E(\mu_t) - E(\tilde{\mu}_t)\| \leq C \cdot \mathcal{W}_p(\mu_t, \tilde{\mu}_t). \tag{63}$$

Second,

$$\left\|E(\tilde{\mu}_t) - E\left(\frac{\tilde{\mu}_t}{\|\tilde{\mu}_t\|_{\mathcal{W}_p}}\right)\right\| \leq C \cdot \mathcal{W}_p(\mu_t, \tilde{\mu}_t) \tag{64}$$

by (57). Lastly,

$$\left\|E\left(\frac{\tilde{\mu}_t}{\|\tilde{\mu}_t\|_{\mathcal{W}_p}}\right) - \|\tilde{\mu}_t\|_{\mathcal{W}_p} E\left(\frac{\tilde{\mu}_t}{\|\tilde{\mu}_t\|_{\mathcal{W}_p}}\right)\right\| = \left(1 - \|\tilde{\mu}_t\|_{\mathcal{W}_p}\right) \cdot \left\|E\left(\frac{\tilde{\mu}_t}{\|\tilde{\mu}_t\|_{\mathcal{W}_p}}\right) - 0\right\|$$

$$= \left(1 - \|\tilde{\mu}_t\|_{\mathcal{W}_p}\right) \cdot \left\|E\left(\frac{\tilde{\mu}_t}{\|\tilde{\mu}_t\|_{\mathcal{W}_p}}\right) - E(\delta_0)\right\|$$

$$\overset{(a)}{\leq} \left(1 - \|\tilde{\mu}_t\|_{\mathcal{W}_p}\right) \cdot C \cdot \mathcal{W}_p\left(\frac{\tilde{\mu}_t}{\|\tilde{\mu}_t\|_{\mathcal{W}_p}}, \delta_0\right) \tag{65}$$

$$\overset{(b)}{=} \left(1 - \|\tilde{\mu}_t\|_{\mathcal{W}_p}\right) \cdot C \cdot \left\|\frac{\tilde{\mu}_t}{\|\tilde{\mu}_t\|_{\mathcal{W}_p}}\right\|_{\mathcal{W}_p}$$

$$= C \cdot \left(1 - \|\tilde{\mu}_t\|_{\mathcal{W}_p}\right) \overset{(c)}{\leq} C \cdot \mathcal{W}_p(\mu_t, \tilde{\mu}_t),$$

where (a) is by (56), (b) is by the definition of $\|\cdot\|_{\mathcal{W}_p}$, and (c) is by (57). Incorporating (63)-(65) into (62) yields

$$\left\| \|\mu_t\|_{\mathcal{W}_p} E\left(\frac{\mu_t}{\|\mu_t\|_{\mathcal{W}_p}}\right) - \|\tilde{\mu}_t\|_{\mathcal{W}_p} E\left(\frac{\tilde{\mu}_t}{\|\tilde{\mu}_t\|_{\mathcal{W}_p}}\right) \right\| \leq 3C \cdot \mathcal{W}_p(\mu_t, \tilde{\mu}_t),$$

which contradicts (61).

Let us now handle the case $L = 0$. For two sequences of numbers $a_t, b_t \in \mathbb{R}$, $t \geq 1$, we say that

$$a_t = o(b_t)$$

if

$$\lim_{t \to \infty} \frac{a_t}{b_t} = 0.$$

Denote

$$d_t := \mathcal{W}_p(\mu_t, \tilde{\mu}_t).$$

According to (59) with $L = 0$, the first component of $\hat{E}(\mu_t) - \hat{E}(\tilde{\mu}_t)$, which equals $\|\mu_t\|_{\mathcal{W}_p} - \|\tilde{\mu}_t\|_{\mathcal{W}_p}$, satisfies

$$\frac{\left| \|\mu_t\|_{\mathcal{W}_p} - \|\tilde{\mu}_t\|_{\mathcal{W}_p} \right|}{\mathcal{W}_p(\mu_t, \tilde{\mu}_t)} \xrightarrow[t \to \infty]{} 0,$$

and thus

$$1 - \|\tilde{\mu}_t\|_{\mathcal{W}_p} = \left| \|\mu_t\|_{\mathcal{W}_p} - \|\tilde{\mu}_t\|_{\mathcal{W}_p} \right| = o(d_t). \tag{66}$$

By the triangle inequality,

$$\|E(\mu_t) - E(\tilde{\mu}_t)\| \leq$$
$$\left\| E(\mu_t) - \|\tilde{\mu}_t\|_{\mathcal{W}_p} E\left(\frac{\tilde{\mu}_t}{\|\tilde{\mu}_t\|_{\mathcal{W}_p}}\right) \right\| + \left\| \|\tilde{\mu}_t\|_{\mathcal{W}_p} E\left(\frac{\tilde{\mu}_t}{\|\tilde{\mu}_t\|_{\mathcal{W}_p}}\right) - E\left(\frac{\tilde{\mu}_t}{\|\tilde{\mu}_t\|_{\mathcal{W}_p}}\right) \right\| \tag{67}$$
$$+ \left\| E\left(\frac{\tilde{\mu}_t}{\|\tilde{\mu}_t\|_{\mathcal{W}_p}}\right) - E(\tilde{\mu}_t) \right\|.$$

We shall show that each of the three above terms is $o(d_t)$.

First, since $\|\mu_t\|_{\mathcal{W}_p} = 1$,

$$\left\| E(\mu_t) - \|\tilde{\mu}_t\|_{\mathcal{W}_p} E\left(\frac{\tilde{\mu}_t}{\|\tilde{\mu}_t\|_{\mathcal{W}_p}}\right) \right\| =$$
$$\left\| \|\mu_t\|_{\mathcal{W}_p} E\left(\frac{\mu_t}{\|\mu_t\|_{\mathcal{W}_p}}\right) - \|\tilde{\mu}_t\|_{\mathcal{W}_p} E\left(\frac{\tilde{\mu}_t}{\|\tilde{\mu}_t\|_{\mathcal{W}_p}}\right) \right\| \overset{(a)}{\leq} \left\| \hat{E}(\mu_t) - \hat{E}(\tilde{\mu}_t) \right\| \overset{(59)}{=} o(d_t), \tag{68}$$

with inequality (a) holding since the argument of the norm on the left consists the last $m$ coordinates of $\hat{E}(\mu_t) - \hat{E}(\tilde{\mu}_t)$. Thus, the first term in (67) is $o(d_t)$. For the second term,

$$
\begin{aligned}
& \left\| \|\tilde{\mu}_t\|_{\mathcal{W}_p} E\left( \frac{\tilde{\mu}_t}{\|\tilde{\mu}_t\|_{\mathcal{W}_p}} \right) - E\left( \frac{\tilde{\mu}_t}{\|\tilde{\mu}_t\|_{\mathcal{W}_p}} \right) \right\| \\
&= \left( 1 - \|\tilde{\mu}_t\|_{\mathcal{W}_p} \right) \cdot \left\| E\left( \frac{\tilde{\mu}_t}{\|\tilde{\mu}_t\|_{\mathcal{W}_p}} \right) \right\| \\
&= \left( 1 - \|\tilde{\mu}_t\|_{\mathcal{W}_p} \right) \cdot \left\| E\left( \frac{\tilde{\mu}_t}{\|\tilde{\mu}_t\|_{\mathcal{W}_p}} \right) - E(\delta_0) \right\| \\
&\overset{(a)}{\leq} \left( 1 - \|\tilde{\mu}_t\|_{\mathcal{W}_p} \right) \cdot C \cdot \mathcal{W}\left( \frac{\tilde{\mu}_t}{\|\tilde{\mu}_t\|_{\mathcal{W}_p}}, \delta_0 \right) \\
&= \left( 1 - \|\tilde{\mu}_t\|_{\mathcal{W}_p} \right) \cdot C \cdot \left\| \frac{\tilde{\mu}_t}{\|\tilde{\mu}_t\|_{\mathcal{W}_p}} \right\|_{\mathcal{W}_p} \\
&= \left( 1 - \|\tilde{\mu}_t\|_{\mathcal{W}_p} \right) C \overset{(b)}{=} o(d_t),
\end{aligned}
\tag{69}
$$

where (a) is by (56) and (b) is by (66).

Finally, by (57),

$$
\left\| E\left( \frac{\tilde{\mu}_t}{\|\tilde{\mu}_t\|_{\mathcal{W}_p}} \right) - E(\tilde{\mu}_t) \right\| \leq C \cdot \left( 1 - \|\tilde{\mu}_t\|_{\mathcal{W}_p} \right) = o(d_t).
\tag{70}
$$

Therefore, by (68)-(70) and (67), we have that

$$
\|E(\mu_t) - E(\tilde{\mu}_t)\| = o(d_t),
$$

and thus $E$ is not lower-Lipschitz. This concludes the proof of Lemma B.11. $\qquad\square$

To finish the proof of Theorem 4.4, suppose that $\Omega \subseteq \mathbb{R}^d$ is an arbitrary set with a nonempty interior. Let $\Omega_0 \subseteq \Omega$ be an open ball contained in $\Omega$, and let $\boldsymbol{x}_0$ be the center of $\Omega_0$. Then $\Omega_0 - \boldsymbol{x}_0$ is an open ball centered at zero.

Given $E : \mathcal{P}_{\leq N}(\Omega) \to \mathbb{R}^m$, define $\tilde{E} : \mathcal{P}_{\leq N}(\Omega_0 - \boldsymbol{x}_0) \to \mathbb{R}^m$ by

$$
\tilde{E}(\mu) := E(\mu + \boldsymbol{x}_0),
$$

where $\mu + \boldsymbol{x}_0$ is the distribution $\mu$ shifted by $\boldsymbol{x}_0$:

$$
\mu + \boldsymbol{x}_0 := \sum_{i=1}^{N} w_i \delta_{\boldsymbol{x}^{(i)} + \boldsymbol{x}_0}.
$$

Then $\tilde{E}$ satisfies the assumptions of Lemma B.11, and thus there exist two sequences of distributions $\mu_t, \tilde{\mu}_t \in \mathcal{P}_{\leq N}(\Omega_0 - \boldsymbol{x}_0)$ such that

$$
\frac{\left\| \tilde{E}(\mu_t) - \tilde{E}(\tilde{\mu}_t) \right\|}{\mathcal{W}_p(\mu_t, \tilde{\mu}_t)} \xrightarrow[t \to \infty]{} L,
$$

with $L = 0$ or $L = \infty$. Note that the sequences of shifted distributions $\{\mu_t + \boldsymbol{x}_0\}_{t \geq 1}$ and $\{\tilde{\mu}_t + \boldsymbol{x}_0\}_{t \geq 1}$ are in $\mathcal{P}_{\leq N}(\Omega_0)$ and thus in $\mathcal{P}_{\leq N}(\Omega)$. Since

$$
\mathcal{W}_p(\mu_t + \boldsymbol{x}_0, \tilde{\mu}_t + \boldsymbol{x}_0) = \mathcal{W}_p(\mu_t, \tilde{\mu}_t),
$$

we have that

$$
\begin{aligned}
\frac{\|E(\mu_t + \boldsymbol{x}_0) - E(\tilde{\mu}_t + \boldsymbol{x}_0)\|}{\mathcal{W}_p(\mu_t + \boldsymbol{x}_0, \tilde{\mu}_t + \boldsymbol{x}_0)} &= \frac{\|E(\mu_t + \boldsymbol{x}_0) - E(\tilde{\mu}_t + \boldsymbol{x}_0)\|}{\mathcal{W}_p(\mu_t, \tilde{\mu}_t)} \\
&= \frac{\left\| \tilde{E}(\mu_t) - \tilde{E}(\tilde{\mu}_t) \right\|}{\mathcal{W}_p(\mu_t, \tilde{\mu}_t)} \xrightarrow[t \to \infty]{} L,
\end{aligned}
$$

which implies that $E$ is not bi-Lipschitz on $\mathcal{P}_{\leq N}(\Omega_0)$, and thus not on $\mathcal{P}_{\leq N}(\Omega)$. $\qquad\square$

We now prove Theorem 4.2. Before proceeding with the proof, we first recall the precise definition of piecewise linearity.

**Definition B.12.** A function $f : \mathbb{R}^m \to \mathbb{R}^n$ is *piecewise linear* if there exists a finite collection of polyhedra[8] $\{P_i\}_{i=1}^k$ that partition $\mathbb{R}^m$ (i.e., their union covers $\mathbb{R}^m$ and their interiors are disjoint) such that $f$ is linear on each $P_i$. That is, for each $i$, there exist a matrix $A_i \in \mathbb{R}^{n \times m}$ and a vector $b_i \in \mathbb{R}^n$ such that

$$f(x) = A_i x + b_i, \quad \text{for all } x \in P_i.$$

**Theorem 4.2.** (Statement in Section 4) *Let* $E : \mathcal{P}_{\leq N}\left(\mathbb{R}^d\right) \to \mathbb{R}^m$ *be injective and positively homogeneous. Let* $\boldsymbol{w}, \tilde{\boldsymbol{w}} \in \Delta^N$ *be fixed, and suppose that the functions* $E(\boldsymbol{X}, \boldsymbol{w})$ *and* $E(\boldsymbol{X}, \tilde{\boldsymbol{w}})$ *are piecewise linear in* $\boldsymbol{X}$. *Then there exist* $C, c > 0$ *such that for all* $\boldsymbol{X}, \tilde{\boldsymbol{X}} \in \mathbb{R}^{d \times N}$ *and* $p \in [1, \infty]$,

$$c \cdot \mathcal{W}_p\Big((\boldsymbol{X}, \boldsymbol{w}), \big(\tilde{\boldsymbol{X}}, \tilde{\boldsymbol{w}}\big)\Big) \leq \left\|E(\boldsymbol{X}, \boldsymbol{w}) - E\big(\tilde{\boldsymbol{X}}, \tilde{\boldsymbol{w}}\big)\right\| \leq C \cdot \mathcal{W}_p\Big((\boldsymbol{X}, \boldsymbol{w}), \big(\tilde{\boldsymbol{X}}, \tilde{\boldsymbol{w}}\big)\Big). \quad (12)$$

*Proof.* The proof is outlined as follows: First we show that there exist constants $\tilde{c}, \tilde{C} > 0$ for which (12) holds in the special case $p = 1$. Then we show that for any fixed $\boldsymbol{w}, \tilde{\boldsymbol{w}} \in \Delta^N$ there exists a constant $\beta > 0$ such that for all $\boldsymbol{X}, \tilde{\boldsymbol{X}} \in \mathbb{R}^{d \times N}$,

$$\mathcal{W}_1\Big((\boldsymbol{X}, \boldsymbol{w}), \big(\tilde{\boldsymbol{X}}, \tilde{\boldsymbol{w}}\big)\Big) \geq \beta \cdot \mathcal{W}_\infty\Big((\boldsymbol{X}, \boldsymbol{w}), \big(\tilde{\boldsymbol{X}}, \tilde{\boldsymbol{w}}\big)\Big). \quad (71)$$

This will imply that for the given pair $\boldsymbol{w}, \tilde{\boldsymbol{w}}$, (12) holds for all $p \in [1, \infty]$ with the constants $c = \tilde{c}\beta$ and $C = \tilde{C}$. To see this, note that by the monotonicity of $\mathcal{W}_p$ with respect to $p$ (Eq. (52)),

$$\mathcal{W}_1\Big((\boldsymbol{X}, \boldsymbol{w}), \big(\tilde{\boldsymbol{X}}, \tilde{\boldsymbol{w}}\big)\Big) \leq \mathcal{W}_p\Big((\boldsymbol{X}, \boldsymbol{w}), \big(\tilde{\boldsymbol{X}}, \tilde{\boldsymbol{w}}\big)\Big) \leq \mathcal{W}_\infty\Big((\boldsymbol{X}, \boldsymbol{w}), \big(\tilde{\boldsymbol{X}}, \tilde{\boldsymbol{w}}\big)\Big). \quad (72)$$

Therefore, Eq. (12) with $p = 1$ and constants $\tilde{C}, \tilde{c}$ and Equations (71) and (72) imply that

$$\begin{aligned}
\left\|E(\boldsymbol{X}, \boldsymbol{w}) - E\big(\tilde{\boldsymbol{X}}, \tilde{\boldsymbol{w}}\big)\right\| &\geq \tilde{c} \cdot \mathcal{W}_1\Big((\boldsymbol{X}, \boldsymbol{w}), \big(\tilde{\boldsymbol{X}}, \tilde{\boldsymbol{w}}\big)\Big) \geq \tilde{c}\beta \cdot \mathcal{W}_\infty\Big((\boldsymbol{X}, \boldsymbol{w}), \big(\tilde{\boldsymbol{X}}, \tilde{\boldsymbol{w}}\big)\Big) \\
&\geq \tilde{c} \cdot \mathcal{W}_p\Big((\boldsymbol{X}, \boldsymbol{w}), \big(\tilde{\boldsymbol{X}}, \tilde{\boldsymbol{w}}\big)\Big)
\end{aligned} \quad (73)$$

and

$$\left\|E(\boldsymbol{X}, \boldsymbol{w}) - E\big(\tilde{\boldsymbol{X}}, \tilde{\boldsymbol{w}}\big)\right\| \leq \tilde{C} \cdot \mathcal{W}_1\Big((\boldsymbol{X}, \boldsymbol{w}), \big(\tilde{\boldsymbol{X}}, \tilde{\boldsymbol{w}}\big)\Big) \leq \tilde{C}\beta \cdot \mathcal{W}_p\Big((\boldsymbol{X}, \boldsymbol{w}), \big(\tilde{\boldsymbol{X}}, \tilde{\boldsymbol{w}}\big)\Big). \quad (74)$$

Let us begin by proving that (12) holds for $p = 1$. The 1-Wasserstein distance between two distributions parametrized by $(\boldsymbol{X}, \boldsymbol{w})$ and $\big(\tilde{\boldsymbol{X}}, \tilde{\boldsymbol{w}}\big)$ can be expressed by

$$\mathcal{W}_1\Big((\boldsymbol{X}, \boldsymbol{w}), \big(\tilde{\boldsymbol{X}}, \tilde{\boldsymbol{w}}\big)\Big) = \min_{\pi \in \Pi(\boldsymbol{w}, \tilde{\boldsymbol{w}})} \sum_{i,j \in [N]} \pi_{ij} \left\|\boldsymbol{x}^{(i)} - \tilde{\boldsymbol{x}}^{(j)}\right\|, \quad (75)$$

where the set $\Pi(\boldsymbol{w}, \tilde{\boldsymbol{w}})$ of transport plans from $(\boldsymbol{X}, \boldsymbol{w})$ to $\big(\tilde{\boldsymbol{X}}, \tilde{\boldsymbol{w}}\big)$ is given by

$$\Pi(\boldsymbol{w}, \tilde{\boldsymbol{w}}) = \left\{ \pi \in [0,1]^{N \times N} \; \middle| \; \forall i \in [N] \sum_{j=1}^N \pi_{ij} = w_i \bigwedge \forall j \in [N] \sum_{i=1}^N \pi_{ij} = \tilde{w}_j \right\}.$$

In particular, $\Pi(\boldsymbol{w}, \tilde{\boldsymbol{w}})$ depends only on $\boldsymbol{w}$ and $\tilde{\boldsymbol{w}}$ and not on the points $\boldsymbol{X}, \tilde{\boldsymbol{X}}$.

Let $\widetilde{\mathcal{W}}_1$ be a modified 1-Wasserstein distance that uses the $\ell_1$-norm rather than $\ell_2$ as its basic cost function:

$$\widetilde{\mathcal{W}}_1\Big((\boldsymbol{X}, \boldsymbol{w}), \big(\tilde{\boldsymbol{X}}, \tilde{\boldsymbol{w}}\big)\Big) := \min_{\pi \in \Pi(\boldsymbol{w}, \tilde{\boldsymbol{w}})} \sum_{i,j \in [N]} \pi_{ij} \left\|\boldsymbol{x}^{(i)} - \tilde{\boldsymbol{x}}^{(j)}\right\|_1. \quad (76)$$

Note that since

$$\|\boldsymbol{x}\|_2 \leq \|\boldsymbol{x}\|_1 \leq \sqrt{d}\|\boldsymbol{x}\|_2 \qquad \forall \boldsymbol{x} \in \mathbb{R}^d, \quad (77)$$

---

[8]A *polyhedron* is a finite intersection of closed half-spaces.

we have
$$\mathcal{W}_1\Big((\boldsymbol{X}, \boldsymbol{w}), \big(\tilde{\boldsymbol{X}}, \tilde{\boldsymbol{w}}\big)\Big) \leq \widetilde{\mathcal{W}}_1\Big((\boldsymbol{X}, \boldsymbol{w}), \big(\tilde{\boldsymbol{X}}, \tilde{\boldsymbol{w}}\big)\Big) \leq \sqrt{d} \cdot \mathcal{W}_1\Big((\boldsymbol{X}, \boldsymbol{w}), \big(\tilde{\boldsymbol{X}}, \tilde{\boldsymbol{w}}\big)\Big). \tag{78}$$

Thus, it suffices to prove an analogue of (12) in which $\mathcal{W}_1$ is replaced by $\widetilde{\mathcal{W}}_1$ and $\|\cdot\|$ by $\|\cdot\|_1$; namely, that for fixed $\boldsymbol{w}, \tilde{\boldsymbol{w}}$, there exist constants $C, c > 0$ such that

$$c \cdot \widetilde{\mathcal{W}}_1\Big((\boldsymbol{X}, \boldsymbol{w}), \big(\tilde{\boldsymbol{X}}, \tilde{\boldsymbol{w}}\big)\Big) \leq \Big\|E(\boldsymbol{X}, \boldsymbol{w}) - E\big(\tilde{\boldsymbol{X}}, \tilde{\boldsymbol{w}}\big)\Big\|_1 \leq C \cdot \widetilde{\mathcal{W}}_1\Big((\boldsymbol{X}, \boldsymbol{w}), \big(\tilde{\boldsymbol{X}}, \tilde{\boldsymbol{w}}\big)\Big). \tag{79}$$

To this end, we define the function $f : \mathbb{R}^{d \times N} \times \mathbb{R}^{d \times N} \to \mathbb{R}^2$,

$$f\big(\boldsymbol{X}, \tilde{\boldsymbol{X}}\big) := \begin{bmatrix} \Big\|E(\boldsymbol{X}, \boldsymbol{w}) - E\big(\tilde{\boldsymbol{X}}, \tilde{\boldsymbol{w}}\big)\Big\|_1 \\ \widetilde{\mathcal{W}}_1\Big((\boldsymbol{X}, \boldsymbol{w}), \big(\tilde{\boldsymbol{X}}, \tilde{\boldsymbol{w}}\big)\Big) \end{bmatrix}.$$

Since both $E$ and $\widetilde{\mathcal{W}}_1$ are positively homogeneous with respect to $\big(\boldsymbol{X}, \tilde{\boldsymbol{X}}\big)$, then so is $f$. Namely,

$$f\big(\alpha \boldsymbol{X}, \alpha \tilde{\boldsymbol{X}}\big) = \alpha f\big(\boldsymbol{X}, \tilde{\boldsymbol{X}}\big), \qquad \forall \alpha \geq 0. \tag{80}$$

In addition to being positively homogeneous, $f$ is also piecewise linear, as we will show below. We then prove that these two properties together imply that $E$ is bi-Lipschitz with respect to $\widetilde{\mathcal{W}}_1$.

Let us now show that $f$ is piecewise linear in $\big(\boldsymbol{X}, \tilde{\boldsymbol{X}}\big)$. The first component of $f$, $\Big\|E(\boldsymbol{X}, \boldsymbol{w}) - E\big(\tilde{\boldsymbol{X}}, \tilde{\boldsymbol{w}}\big)\Big\|_1$, is clearly piecewise linear, as it is the composition of the $\ell_1$-norm with a piecewise-linear function. We shall now show that the second component $\widetilde{\mathcal{W}}_1\Big((\boldsymbol{X}, \boldsymbol{w}), \big(\tilde{\boldsymbol{X}}, \tilde{\boldsymbol{w}}\big)\Big)$ is also piecewise linear. For any fixed $\boldsymbol{X}$ and $\tilde{\boldsymbol{X}}$, the optimization problem in (76) is a linear program in $\pi$, with the set of feasible solutions being the compact polyhedron $\Pi(\boldsymbol{w}, \tilde{\boldsymbol{w}})$. Thus, the optimal solution must be attained at one of the vertices of $\Pi(\boldsymbol{w}, \tilde{\boldsymbol{w}})$. As any polyhedron has a finite number of vertices[9], let $\pi^{(1)}, \dots, \pi^{(K)}$ be the vertices of $\Pi(\boldsymbol{w}, \tilde{\boldsymbol{w}})$, and recall that these vertices do not depend on $\big(\boldsymbol{X}, \tilde{\boldsymbol{X}}\big)$. Therefore, (76) can be reformulated as

$$\widetilde{\mathcal{W}}_1\Big((\boldsymbol{X}, \boldsymbol{w}), \big(\tilde{\boldsymbol{X}}, \tilde{\boldsymbol{w}}\big)\Big) = \min_{k \in [K]} \sum_{i,j \in [N]} \pi_{ij}^{(k)} \Big\|\boldsymbol{x}^{(i)} - \tilde{\boldsymbol{x}}^{(j)}\Big\|_1. \tag{81}$$

From (81) it can be seen that $\widetilde{\mathcal{W}}_1\Big((\boldsymbol{X}, \boldsymbol{w}), \big(\tilde{\boldsymbol{X}}, \tilde{\boldsymbol{w}}\big)\Big)$ is piecewise linear in $\big(\boldsymbol{X}, \tilde{\boldsymbol{X}}\big)$, as it is the minimum of a finite number of piecewise-linear functions. Since the concatenation of piecewise-linear functions is also piecewise linear, we have that $f\big(\boldsymbol{X}, \tilde{\boldsymbol{X}}\big)$ is piecewise linear.

Now, let $A \subseteq \mathbb{R}^2$ be the image of $f$:

$$A := \Big\{ f\big(\boldsymbol{X}, \tilde{\boldsymbol{X}}\big) \ \Big| \ \boldsymbol{X}, \tilde{\boldsymbol{X}} \in \mathbb{R}^{d \times N} \Big\}.$$

Since $f$ is piecewise linear, it maps the space $\mathbb{R}^{d \times N} \times \mathbb{R}^{d \times N}$ to a finite union of closed polyhedra (some of which may be unbounded). Hence, $A$ is a finite union of closed sets, and thus is closed.

Denote the two coordinates of $f$ by

$$f_1\big(\boldsymbol{X}, \tilde{\boldsymbol{X}}\big) = \Big\|E(\boldsymbol{X}, \boldsymbol{w}) - E\big(\tilde{\boldsymbol{X}}, \tilde{\boldsymbol{w}}\big)\Big\|_1,$$
$$f_2\big(\boldsymbol{X}, \tilde{\boldsymbol{X}}\big) = \widetilde{\mathcal{W}}_1\Big((\boldsymbol{X}, \boldsymbol{w}), \big(\tilde{\boldsymbol{X}}, \tilde{\boldsymbol{w}}\big)\Big).$$

Since $E$ is injective by assumption, if $f_1\big(\boldsymbol{X}, \tilde{\boldsymbol{X}}\big) = 0$ then $f_2\big(\boldsymbol{X}, \tilde{\boldsymbol{X}}\big) = 0$. Thus, $A$ does not contain the point $(0, 1)$. Conversely, if $f_2\big(\boldsymbol{X}, \tilde{\boldsymbol{X}}\big) = 0$, then $(\boldsymbol{X}, \boldsymbol{w})$ and $\big(\tilde{\boldsymbol{X}}, \tilde{\boldsymbol{w}}\big)$ represent the same distribution, that is,

$$\sum_{i=1}^{N} w_i \delta_{\boldsymbol{x}^{(i)}} = \sum_{i=1}^{N} \tilde{w}_i \delta_{\tilde{\boldsymbol{x}}^{(i)}},$$

---

[9]See (Grünbaum, 2003), Theorem 3, page 32, and the definition of polyhedral sets on page 26 therein.

and since $E(\boldsymbol{X}, \boldsymbol{w}) : \mathcal{P}_{\leq N}(\mathbb{R}^d) \to \mathbb{R}^m$ depends only on the input distribution and not on its particular representation, we have that $f_1\left(\boldsymbol{X}, \tilde{\boldsymbol{X}}\right) = 0$. Hence, $A$ does not contain the point $(1, 0)$.

Now, suppose by contradiction that there do not exist constants $C, c > 0$ for which (79) holds. Then, there exist two sequences $\boldsymbol{X}_t, \tilde{\boldsymbol{X}}_t \in \mathbb{R}^{d \times N}$ such that both coordinates of $f\left(\boldsymbol{X}_t, \tilde{\boldsymbol{X}}_t\right)$ are nonzero for all $t \in \mathbb{N}$, and their ratio satisfies

$$\frac{f_1\left(\boldsymbol{X}_t, \tilde{\boldsymbol{X}}_t\right)}{f_2\left(\boldsymbol{X}_t, \tilde{\boldsymbol{X}}_t\right)} \to L, \tag{82}$$

where the limit $L$ is either $0$ or $\infty$.

Suppose that $L = \infty$. By the positive homogeneity of $E$ and $\widetilde{\mathcal{W}}_1$, we can divide both $\boldsymbol{X}_t$ and $\tilde{\boldsymbol{X}}_t$ by $f_1\left(\boldsymbol{X}_t, \tilde{\boldsymbol{X}}_t\right)$ while still satisfying (82) with $L = \infty$. With these new sequences, we obtain

$$f_1\left(\boldsymbol{X}_t, \tilde{\boldsymbol{X}}_t\right) = 1 \quad \forall t \in \mathbb{N},$$

$$f_2\left(\boldsymbol{X}_t, \tilde{\boldsymbol{X}}_t\right) \xrightarrow[t \to \infty]{} 0.$$

It follows that there exists a sequence in $A$ that converges to the point $(1, 0)$. Since $A$ is closed, we conclude that $(1, 0) \in A$, contradicting our assumption.

By a similar argument, if $L = 0$, then $(0, 1) \in A$, leading to the same contradiction.

To finish the proof, it is left to show that (71) holds with some constant $\beta > 0$ assuming that $\boldsymbol{w}$ and $\tilde{\boldsymbol{w}}$ are fixed. To this end, define the sets $I_k \subseteq [N]^2$ for $k \in [K]$, with $K$ as in (71),

$$I_k := \left\{ (i, j) \in [N]^2 \;\middle|\; \pi_{ij}^{(k)} > 0 \right\}.$$

By the definition of $\Pi(\boldsymbol{w}, \tilde{\boldsymbol{w}})$, for all $k \in [K]$, $I_k$ is nonempty. Let

$$\epsilon_k := \min_{(i,j) \in I_k} \pi_{ij}^{(k)}, \quad k \in [K]. \tag{83}$$

That is, $\epsilon_k$ is the smallest positive entry of $\pi_{ij}^{(k)}$, and by the above, indeed $\epsilon_k > 0$. Let

$$\epsilon_{\min} := \min_{k \in [K]} \epsilon_k > 0. \tag{84}$$

Thus, for all $\boldsymbol{X}, \tilde{\boldsymbol{X}} \in \mathbb{R}^{d \times N}$,

$$\sqrt{d} \cdot \mathcal{W}_1\left((\boldsymbol{X}, \boldsymbol{w}), \left(\tilde{\boldsymbol{X}}, \tilde{\boldsymbol{w}}\right)\right) \stackrel{(78)}{\geq} \widetilde{\mathcal{W}}_1\left((\boldsymbol{X}, \boldsymbol{w}), \left(\tilde{\boldsymbol{X}}, \tilde{\boldsymbol{w}}\right)\right)$$

$$\stackrel{(81)}{=} \min_{k \in [K]} \sum_{i,j \in [N]} \pi_{ij}^{(k)} \left\|\boldsymbol{x}^{(i)} - \tilde{\boldsymbol{x}}^{(j)}\right\|_1 \stackrel{(77)}{\geq} \min_{k \in [K]} \sum_{i,j \in [N]} \pi_{ij}^{(k)} \left\|\boldsymbol{x}^{(i)} - \tilde{\boldsymbol{x}}^{(j)}\right\|_2$$

$$\stackrel{(a)}{=} \min_{k \in [K]} \sum_{(i,j) \in I_k} \pi_{ij}^{(k)} \left\|\boldsymbol{x}^{(i)} - \tilde{\boldsymbol{x}}^{(j)}\right\|_2 \stackrel{(83)}{\geq} \min_{k \in [K]} \sum_{(i,j) \in I_k} \epsilon_k \left\|\boldsymbol{x}^{(i)} - \tilde{\boldsymbol{x}}^{(j)}\right\|_2$$

$$\stackrel{(84)}{\geq} \min_{k \in [K]} \sum_{(i,j) \in I_k} \epsilon_{\min} \left\|\boldsymbol{x}^{(i)} - \tilde{\boldsymbol{x}}^{(j)}\right\|_2 \geq \min_{k \in [K]} \max_{(i,j) \in I_k} \epsilon_{\min} \left\|\boldsymbol{x}^{(i)} - \tilde{\boldsymbol{x}}^{(j)}\right\|_2$$

$$\stackrel{(b)}{=} \epsilon_{\min} \cdot \min_{k \in [K]} \max\left\{ \left\|\boldsymbol{x}^{(i)} - \tilde{\boldsymbol{x}}^{(j)}\right\|_2 \;\middle|\; ij \in [N], \pi_{ij}^{(k)} > 0 \right\}$$

$$\stackrel{(c)}{=} \epsilon_{\min} \cdot \min_{\pi \in \left\{\pi^{(k)}\right\}_{k=1}^{[K]}} \max\left\{ \left\|\boldsymbol{x}^{(i)} - \tilde{\boldsymbol{x}}^{(j)}\right\|_2 \;\middle|\; ij \in [N], \pi_{ij} > 0 \right\}$$

$$\stackrel{(d)}{\geq} \epsilon_{\min} \cdot \min_{\pi \in \Pi(\boldsymbol{w}, \tilde{\boldsymbol{w}})} \max\left\{ \left\|\boldsymbol{x}^{(i)} - \tilde{\boldsymbol{x}}^{(j)}\right\|_2 \;\middle|\; ij \in [N], \pi_{ij} > 0 \right\}$$

$$\stackrel{(e)}{=} \epsilon_{\min} \cdot \mathcal{W}_\infty\left((\boldsymbol{X}, \boldsymbol{w}), \left(\tilde{\boldsymbol{X}}, \tilde{\boldsymbol{w}}\right)\right).$$

where (a) is since $\pi_{ij}^{(k)} = 0$ whenever $(i, j) \notin I_k$; (b) is by the definition of $I_k$; (c) is a simple reformulation; (d) is since the minimum is taken over a larger set $\Pi(\boldsymbol{w}, \tilde{\boldsymbol{w}}) \supseteq \left\{\pi^{(k)}\right\}_{k=1}^{[K]}$; and (e) is by the definition of $\mathcal{W}_\infty$. Hence, (71) holds with $\beta = \frac{\epsilon_{\min}}{\sqrt{d}}$ and the theorem is proven. $\qquad\square$

## C    EXPERIMENT DETAILS

**Hardware**    All experiments were conducted on a single Nvidia A40 GPU.

### C.1    COMPARISON WITH PSWE

In our comparison with PSWE, shown in Table 1, we used pairs of multisets of the form $\boldsymbol{X} = \left\{ \frac{i}{n+1} \mathbf{1} \right\}_{i=1}^{n}$ for two different values of $n$, denoted $n_1, n_2$, where $\mathbf{1} \in \mathbb{R}^d$ is the vector whose entries all equal 1. For example, one could choose

$$\boldsymbol{X}_1 = \left\{ \tfrac{1}{3}, \tfrac{2}{3} \right\}, \quad \boldsymbol{X}_2 = \left\{ \tfrac{1}{4}, \tfrac{2}{4}, \tfrac{3}{4} \right\}.$$

Despite the fact that these multisets are distinct, the interpolation used by PSWE (see e.g., Naderial-izadeh et al. (2021)) identifies these two multisets with the same continuous distribution, and hence assigns them the same embedding. In Table 1 we used $d = 3$ and $n_1 = 5$ and $n_2 = 200$, and found, as expected, that PSWE assigns the same value to both multisets, up to numerical error, regardless of the embedding dimension $m$. In contrast, our method successfully separates $\boldsymbol{X}_1$ and $\boldsymbol{X}_2$ even with $m = 1$, and yields a good approximation of the sliced Wasserstein distance as $m$ increases. Similar results to those shown in the table were obtained for other choices of $d, n_1, n_2$.

Note that computing the sliced Wasserstein distance between such pairs is straightforward, since, for any two multisets $\boldsymbol{X}_1, \boldsymbol{X}_2$ in $\mathbb{R}^d$ with all points belonging to the same line, it can be shown that

$$\mathcal{SW}(\boldsymbol{X}_1, \boldsymbol{X}_2) = \frac{1}{\sqrt{d}} \mathcal{W}(\boldsymbol{X}_1, \boldsymbol{X}_2),$$

and $\mathcal{W}(\boldsymbol{X}_1, \boldsymbol{X}_2)$ can be computed using a linear programming solver.

The PSWE method takes two input parameters—the number of slices $L$ and embedding dimension $N$, which, unlike our method, need not be identical. We set both parameters to the embedding dimension $m$.

### C.2    LEARNING TO APPROXIMATE THE WASSERSTEIN DISTANCE

In this experiment we used embedding dimensions $m_1 = m_2 = 1000$. The MLP consisted of three layers with a hidden dimension of 1000. With this choice of hyperparameters, our model has roughly 3 million learnable parameters and 5 million parameters in total. These hyperparameters were picked manually. The performance of our architecture did not exhibit high sensitivity to the choice of hyperparameters: on most datasets, similar results were obtained with MLPs consisting of 2 to 8 layers, and with hidden dimensions of 500, 1000, 2000 and 4000.

We used fixed parameters for the first embedding $E_1$ and learnable parameters for the second embedding $E_2$. This choice was made since $E_1$ is, in most cases, supposed to handle arbitrary input point clouds, whereas the input to $E_2$ is more specific, in that it is always a set of two vectors that are outputs of $E_1$. Thus, in principle the architecture may benefit from tuning $E_2$ to its particular input structure. In practice, using fixed parameters in both embeddings did not significantly impair performance.

Remarkably, applying an MLP to the input points prior to embedding them via $E_1$ (i.e. adding a feature transform), as well as applying an MLP to the two outputs of $E_1$ prior to embedding them via $E_2$, *impaired* rather than improved the performance. This indicates that our embedding is expressive enough to encode all the required information from the input multisets in a way that facilitates processing by the MLP $\Phi$, thus making additional processing at intermediate steps unnecessary.

Inference times for one pair of multisets were less than half a second for the `ModelNet-large` dataset, and less than 0.2 seconds for the rest of the datasets. The training times of the competing models appear in Table 3.

Training was performed on an Nvidia A40 GPU, whereas the rest of the methods were trained over an Nvidia RTX A6000 GPU, both of which have similar performance on 32-bit floating point (37.4 and 38.7 TFLOPS).

Exact computation of the 1-Wasserstein distance using the `ot.emd2()` function of the Python Optimal Transport package (Flamary et al., 2021) was up to 2.5 times slower than our method (2 to

5 ms vs 1.9 ms) on small multisets (less than 300 elements) and 150 times slower (640 ms vs 4.2 ms) on large multisets (`ModelNet-large`).

## C.3 ROBUSTNESS TO PARAMETER REDUCTION

In both architectures, the input and output dimension of each MLP, as well as that of the FSW embedding, were multiplied by a scaling factor to obtain the desired number of parameters. Training our model in all problem instances took between 60 to 65 minutes. Training PointNet took between 4:43 hours to 5 hours, and was done using the original code of (Xia, 2019).

