# OpenReview forum: "Fourier Sliced-Wasserstein Embedding for Multisets and Measures"
_ICLR.cc/2025/Conference — ICLR 2025 Poster_

### Official Review · Reviewer_styv · 2024-10-27

**Soundness:** 2
**Presentation:** 3
**Contribution:** 2
**Rating:** 6
**Confidence:** 4

**Summary:**

**Summary:**
The paper introduces the "Fourier Sliced Wasserstein (FSW) embedding" for data in \(\mathbb{R}^d\).

**Theoretical Contributions:**
1. The authors prove that the embedding preserves or approximates the sliced Wasserstein distance.
2. They also demonstrate that the embedding technique is injective and bi-Lipschitz.

**Numerical Experiments:**
1. The authors evaluate the approximation error of the proposed Fourier Sliced Wasserstein embedding.
2. They showcase an application of FSW for approximating the Wasserstein distance using a Multi-Layer Perceptron (MLP).

**Strengths:**

1. The combination of the Fourier/cosine transform and the sliced Wasserstein distance (see Eq. (6)) is a novel approach.
2. Theoretical properties for this new technique with respect to the uniform distribution, along with its empirical approximation, are proposed (see Theorem 3.2, Corollary 3.3).
3. Injectivity and bi-Lipschitz properties of the embedding have been investigated.

**Weaknesses:**

1. I recommend adding a section to introduce baseline methods. For example, explaining how Sinkhorn [Cuturi, 2013] can be used to train a neural network as a Wasserstein distance estimator. Currently, the experimental setup (E1, E2, Phi, Leaky-ReLU) appears tailored only to the proposed method in this paper.
2. It would be beneficial to introduce a real-data application of the proposed Sliced Wasserstein distance embedding technique to illustrate its practical utility.
3. I’m unclear on why 'bi-Lipschitz' is considered a crucial property. Could you provide an example to clarify? For instance, in which applications would the lack of a bi-Lipschitz property cause issues, and where having this property could offer distinct advantages?

**Questions:**

1. Regarding point (2), is "Multisets" simply another term for "discrete distributions"?
2. Could you clarify what "E2" refers to in lines 473-474?

---

> ### Author Response · Authors · 2024-11-14
> **Response to Reviewer styv (Part 1 of 2)**
>
> **Response to summary:**
>
> We would like to highlight that, in addition to the theoretical guarantees for our embedding, we present the impossibility result stated in Theorem 4.4. This result proves that it is impossible to embed discrete distributions into any finite-dimensional Euclidean space in a bi-Lipschitz manner. This saves the community further efforts in that direction, and essesntially shows that an embedding with substantially better analytical properties than the FSW does not exist.
>
>
>
> **Weakness 1: Introducing baseling methods**
>
> Thank you for your suggestion. We added to the revised manuscript a paragraph describing all the baseline methods (lines 498-504).
>
> We would like to stress that the methods presented in Table 2 are those introduced in [Chen] and earlier papers, and were not implemented by us. Specifically: $\mathcal{N}\_{\textup{ProductNet}}$, $\mathcal{N}\_{\textup{SDeepSets}}$ and WPCE use their own architectures, and Sinkhorn is an approximation algorithm specifically designed to approximate the $p$-Wasserstein distance.
>
> Our architecture based on $E_1$, $E_2$, $\Phi$, described in line 485,  achieves state of the art results with a simple combination of our embedding and one MLP. In comparison, $\mathcal{N}{\_\textup{ProductNet}}$ produces inferior results using three MLPs.
>
> The only method other than ours that we tested with our architecture was PSWE, which is designed to compute Sliced-Wasserstein preserving embeddings.
>
> Lastly, the reason why Sinkhorn cannot be incorporated into our architecture is due to its own inherent limitation: it takes _pairs_ of input distributions and estimates their distances, rather than computing a distance-preserving embedding for individual distributions. This significantly limits its applicability to practical learning tasks, as we further discuss in our response to Reviewer Ew1o. We also added a brief explanation in line 307.
>
> **Weakness 2: Real-data application of our method**
>
> The experiment presented in Table 2 illustrates the utility of our embedding in a learning task on real-world data (ModelNet-40). We note that our paper includes all experiments from the NeurIPS-accepted work by Chen and Wang [Chen], as well as theoretical results that in our opinion merit acceptance in their own right.
>
> _To be continued..._

---

> ### Author Response · Authors · 2024-11-14
> **Response to Reviewer styv (Part 2 of 2)**
>
> **Weakness 3: Why bi-Lipschitzness is important**
>
> The lack of bi-Lipschitzness, which plagues most prevalent multiset architectures to date, practically implies that there inevitably exist pairs of different input multisets that appear numerically identical to the architecture. This poses a problem, for example, in classification tasks where such pairs need to be assigned different labels. Any multiset architecture based on sum- or max-pooling is provably affected by this problem [FWT].
>
> Achieving a bi-Lipschitz embedding for multisets has been recognized as an important goal in previous works. Our work is the first to fully achieve this goal. Below is a selection of previous works that underscore the importance of bi-Lipschitzness:
>
> > "The question of which metric spaces admit a bilipschitz embedding into some (finite-dimensional) Euclidean space is natural, and has received a lot of attention in recent work. The results obtained so far indicate that there is no simple answer to this question."
> >
> > — Lang, Urs, and Conrad Plaut. "Bilipschitz embeddings of metric spaces into space forms." _Geometriae Dedicata 87_ (2001): 285-307.
>
>
> > "Since the late 1990’s, it has become apparent that designing efficient approximate nearest neighbor algorithms, at least for high-dimensional data, is closely related to the task of designing _low-distortion embeddings_. A _bi-Lipschitz embedding_ between two metric spaces $(X,d\_X)$ $(X',d\_{X'})$ is a mapping $f : X \to X'$ such that ... where the parameter $D \geq 1$ called [_sic_] the _distortion_ of $f$."
> >
> > — Indyk, Piotr, and Assaf Naor. "Nearest-neighbor-preserving embeddings." _ACM Transactions on Algorithms (TALG)_ 3.3 (2007): 31-es.
>
> > "The second negative result is that while moments of MLPs with analytic activations can be injective, they can never be stable in the bi-Lipschitz sense. This points to a possible advantage of injective multiset functions that are not based on moments, but rather on sorting or max-filters."
> >
> > — Amir, T., Gortler, S., Avni, I., Ravina, R., & Dym, N. (2023). "Neural injective functions for multisets, measures and graphs via a finite witness theorem." _Advances in Neural Information Processing Systems (NeurIPS)_ 37 (2023)
>
>
> > “We propose developing fine-grained expressivity results, namely metric equivalencies between explicit graph metrics and feature metrics for GNNs on graphs with features. An ideal result would derive a bi-Lipschitz correspondence between such metrics.”
> >
> > — Christopher Morris, Nadav Dym, Haggai Maron, İsmail İlkan Ceylan, Fabrizio Frasca, Ron Levie, Derek Lim, Michael Bronstein, Martin Grohe, and Stefanie Jegelka. "Future Directions in Foundations of Graph Machine Learning." _International Conference on Machine Learning (ICML)_ (2024)
>
> **Question 1: Regarding point (2), is "Multisets" simply another term for "discrete distributions"?**
>
> _Multisets_ and _discrete distributions_ refer to different concepts. Multisets are essentially sets that account for repetitions. For instance, $\\{b,a,b\\} = \\{a,b,b\\} \neq \\{a,b\\}$. Discrete distributions, on the other hand, are probability distributions with finite support. However, multisets can be idenitified with the subset of of discrete distributions with uniform weights, as discussed beginning at line 183.
>
> **Question 2: Could you clarify what "E2" refers to in lines 473-474?**
>
> $E_1$ and $E_2$ are two independent instances of the FSW embedding, with different input and output dimensions. $E_1$ maps distributions over $\mathbb{R}^d$ into $\mathbb{R}^{m_1}$, and $E_2$ maps distributions over $\mathbb{R}^{m_1}$ into $\mathbb{R}^{m_2}$, with $m_1$, $m_2$ being architecture hyperparameters. This is detailed in the manuscript (lines 481-485 in the revised version).
>
> **References:**
>
> [Chen] Chen, S., Wang, Y. "Neural approximation of Wasserstein distance via a universal architecture for symmetric and factorwise group invariant functions." _Advances in Neural Information Processing Systems (NeurIPS)_ 37 (2023).
>
> [FWT] Amir, T., Gortler, S., Avni, I., Ravina, R., & Dym, N. (2023). "Neural injective functions for multisets, measures and graphs via a finite witness theorem." _Advances in Neural Information Processing Systems (NeurIPS)_ 37 (2023).

---

> > ### Comment · Reviewer_styv · 2024-12-02
> >
> > The reviewer thank the authors' response. I will keep my rate (accept).

---

### Official Review · Reviewer_gmod · 2024-11-02

**Soundness:** 4
**Presentation:** 4
**Contribution:** 3
**Rating:** 8
**Confidence:** 3

**Summary:**

The paper seeks to establish a mapping from multisets and measures over $ \mathbb{R}^d $ into Euclidean space, ensuring that the sliced Wasserstein distance corresponds to the distance between their mappings in the target space. The authors propose a mapping that is bi-Lipschitz for multisets and injective for measures. Additionally, they demonstrate that a bi-Lipschitz map for measures does not exist.

**Strengths:**

The paper is well-structured, and its message is clear. The proofs provided are rigorous and exceptionally clear. This particular problem is quite interesting. I really enjoyed reading the paper.

**Weaknesses:**

I don't see any weaknesses. Therefore, I recommend accepting it.

**Questions:**

I haven't any question.

---

> ### Author Response · Authors · 2024-11-14
> **Response to Reviewer gmod**
>
> Thank you :)

---

### Official Review · Reviewer_kJBi · 2024-11-03

**Soundness:** 3
**Presentation:** 3
**Contribution:** 3
**Rating:** 5
**Confidence:** 2

**Summary:**

This paper presents a novel approach to high-dimensional dataset embedding. The authors provided theoretical performance guarantees and numerical study to show the superior performance of the framework.

**Strengths:**

The theoretical contribution seems to be sound, with explicitly stated technical assumptions and results. Numerical study is solid.

**Weaknesses:**

1. The authors denovted much space to describe the p-wasserstein and infinity-type Wasserstein distance. Why it is necessary to introduce infinity-type Wasserstein distance?
2. In line 222, the authors mentioned that in the special case of d=1, Wasserstien can be computed significantly fast. So what is the complexity rate?
3. In line 344, what is the definition of STD???
4. The authors should provide proof ideas for the main technical results in the main content.

**Questions:**

I am new to this field. Could the authors elaborate more on the practical motivation and applications of this approach?

---

> ### Author Response · Authors · 2024-11-14
> **Response to Reviewer kJBi**
>
> We thank the reviewer for the comments. We have uploaded a revised version of the manuscript, where the comments have been addressed and incorporated.
>
> **Response to summary:**
>
> We would like to highlight an additional theoretical contribution in our paper: the impossibility result of Theorem 4.4, which proves that it is impossible to embed discrete distributions into any finite-dimensional Euclidean space in a bi-Lipschitz manner. This saves the community further efforts in that direction, and essesntially shows that an embedding with substantially better analytical properties than the FSW does not exist.
>
> **Response to weaknesses:**
>
> 1. **On the inclusion of $\mathcal{W}_{\infty}$:** We included the definition of the $p$-Wasserstein distance with $p = \infty$ to allow our results to be stated across all $p \in [0, \infty]$. Our bi-Lipschitzness guarantee and impossibility result apply uniformly for all $p$ in this range.
> 2. **Complexity of Wasserstein when $d=1$**: The complexity in this case is the computational complexity of the sort function, which is $\mathcal{O}(n \log n)$. This is stated in lines 223-224.
> 3. **Definition of STD:** STD here is the Standard Deviation. We clarified this in the revised manuscript, l. 341-342.
> 4. **Proof ideas in the main text:** We added an overview of the proof ideas of Theorems 4.1, 4.2 and 4.4 to the revised manuscript. Thank you for this comment.
>
> **Response to question on the practical motivation:**
>
> To illustrate the advantage of our approach for practical applications, consider a learning task on multisets handled by traditional architectures based on sum- or max-pooling. With these methods, certain pairs of input multisets may appear numerically identical, meaning the architecture will not be able to distinguish between them—even if they represent different underlying data. In contrast, our approach, due to its bi-Lipschitzness guarantee, can distinguish between these multisets in a way that reflects their actual differences. This practical advantage is evidenced in our experimental results shown in Table 2.

---

> > ### Author Response · Authors · 2024-12-01
> > **Follow Up**
> >
> > Dear Reviewer kJBi,
> >
> > We wanted to follow up regarding our response to your review. In our rebuttal, we addressed all the points raised, including the practical motivation for our approach, proof ideas, and other requested clarifications.
> >
> > If you find that our revision and explanations have resolved your concerns, we would greatly appreciate it if you could consider revisiting and potentially adjusting your score.
> >
> > Thank you for your review and feedback. Please don’t hesitate to reach out if there are any remaining points we can clarify further.
> >
> > Best regards,
> > The Authors

---

### Official Review · Reviewer_Ew1o · 2024-11-05

**Soundness:** 2
**Presentation:** 3
**Contribution:** 2
**Rating:** 6
**Confidence:** 4

**Summary:**

This paper considers Fourier slicing embedding both for a collection of probability distributions and multisets over $\mathbb{R}^d$ and supported at $n$ points. The embedding consists of a projection sample on a 1-dimensional vector on the sphere then calculates a cosine transform of the projected quantile function. Under a specific probability distribution of the frequency, the authors prove that the expectation of the estimation error between the embedded measures is exactly the sliced Wasserstein distance. A second part of the theoretical results consists of proving the injectivity of the embedding under the assumption that the dimension embedding $m \geq 2n(d+1) +1$. Numerical experiments are conducted on point cloud classification.

**Strengths:**

- The paper is well-written and easy to follow. Proofs are rigorous.
- Proposing the sliced embedding Wasserstein (SEW) through a cosine transform of the projected quantile function. Sampling the quantile function via cosine transform is novel.
- Injectivity and bi-Lipschitz properties of FSEW on the collection of multisets.
- Numerical experiments showcase better Wasserstein approximation on simulated datasets and three real datasets than NProductNet, WPCE, NSDeepSets, and Sinkhorn.

**Weaknesses:**

- Several approaches for the derivative of sliced Wasserstein distance like, distributional sliced Wasserstein (Nguen et al, ICLR'21), max-sliced Wasserstein, etc ... Could you highlight the difference between FSW and the SOTA derivative of sliced Wasserstein?

**Questions:**

See Weaknes section.

---

> ### Author Response · Authors · 2024-11-13
> **Response to Reviewer Ew1o**
>
> There is a fundamental difference between our embedding approach and approaches such as Distributional Sliced Wasserstein and Max-Sliced Wasserstein. Our approach takes one input distribution at a time and computes an _embedding_, whereas the aforementioned approaches take two input distributions and estimate a _distance_ between them. Pairwise methods have two disadvantages in comparison with embeddings: (i) higher computational complexity when computing multiple pairwise disances, and (ii) limited applicability to real-world learning problems.
>
> In terms of applicability, a pairwise method cannot be directly applied to common learning tasks, such as object classification, where the inputs are typically individual distributions. In contrast, an embedding is readily applicable to such tasks, as demonstrated in our experiments.
>
> In terms of computation, pairwise methods to estimate sliced optimal transport distances typically require $\tilde{\mathcal{O}}(mnd)$ time (neglecting logarithmic factors), where $m$ is the number of slices, $n$ is the maximal number of support points, and $d$ is the ambient dimension of the support. Thus, computing all pairwise distances for a set of $k$ distributions would take $\tilde{\mathcal{O}}(k^2 mnd)$ time. In contrast, computing our embedding takes $\tilde{\mathcal{O}}(mnd)$ time for each input distribution, and pairwise distances can then be computed in the Euclidean space $\mathbb{R}^m$, resulting in a considerably lower total complexity of $\tilde{\mathcal{O}}(k mnd + k^2 m)$. This approach is therefore significantly more scalable for large datasets where pairwise distance computations are required.
>
> We appreciate the reviewer's comment and will clarify this in the paper.

---

> > ### Comment · Reviewer_Ew1o · 2024-11-26
> >
> > I thank the authors for their answers to my concern. I am keeping my score the same.

---

### Author Response · Authors · 2024-11-16
**Follow Up**

Dear Reviewers,

Thank you for your thoughtful comments on our submission.

We have responded to all of your comments and uploaded a revised version of the manuscript, carefully addressing your feedback. Changes are highlighted in blue for your convenience.

We would greatly appreciate it if you could confirm whether our responses and the revised manuscript adequately address your concerns. This will help us ensure that we have addressed all your feedback within the discussion period.

Thank you for your time and consideration.

Best regards,
The Authors

---

### Author Response · Authors · 2024-11-24
**Reminder**

Dear Reviewers,

Thank you again for your thoughtful comments on our submission. As mentioned in our previous post,
we have responded to all of your comments and uploaded a revised version of the manuscript, carefully addressing your feedback. Changes are highlighted in blue for your convenience.

As the discussion period comes to a close, we would greatly appreciate it if you could confirm whether our responses and the revised manuscript adequately address your concerns.

Thank you for your time and consideration.

Best regards,

The Authors

---

### Meta-Review · Area_Chair_oTZQ · 2024-12-21

**Metareview:**

This paper proposes a novel approach to high-dimensional dataset embedding. Most reviewers found
that the paper is of interest and provide relevant contributions to the field.

**Additional Comments On Reviewer Discussion:**

They were few discussions beyond the rebuttals as most authors are happy about the paper.

---

### Decision · Program_Chairs · 2025-01-22

Accept (Poster)